# How Do Repeated Viewings in Forest Landscapes Influence Young People’s Visual Behaviors and Cognitive Evaluations?

**DOI:** 10.3390/ijerph20064753

**Published:** 2023-03-08

**Authors:** Mengyun Wu, Yu Gao, Zhi Zhang, Bo Zhang, Huan Meng, Weikang Zhang, Tong Zhang

**Affiliations:** 1Landscape Planning Laboratory, Forestry College, Shenyang Agricultural University, Shenyang 110161, China; 2Shenyang Construction Engineering Project Management Center Co., Ltd., Shenyang 110032, China

**Keywords:** visual behavior, forest environment, repeated viewings, cross-recurrence analysis, psychological evaluation

## Abstract

Background: With the spread of the COVID-19 epidemic, it has gradually become normal to periodically visit and enjoy forest landscape resources in the suburbs of cities. For designers and managers of forest landscapes, exploring change in the visual behaviors and cognitive evaluations of people who repeatedly view forest landscapes and the characteristics of this change will aid the design and sustainable utilization of forest landscape resources in the suburbs of cities. Purpose: From the perspective of users’ preferences for forest landscape space, this study explored the changes in visual behavior characteristics and psychological preference characteristics for individuals who repeatedly view forest landscapes and their drivers under different preferences. Methods: This study collected data from 52 graduate and undergraduate students. We used a difference test to compare the differences in the visual behavior coincidence degree and the changes in psychological evaluations; a descriptive statistical analysis to explore young peoples’ likes and dislikes of landscape elements; and Spearman correlation analysis to explore the correlation between the psychological evaluations and visual behaviors. Main results: 1. At the second viewing, the participants’ regression behavior tended to decrease for various spaces, and they were more inclined to view areas that they had not viewed before. In addition, at the second viewing, the degree of fixation behavior coincidence was generally low, and there were obvious differences across spaces; 2. The participants’ feature evaluations and comprehensive evaluations for landscapes did not change significantly with their increased familiarity with the spaces; 3. There was a significant positive correlation between the participants’ psychological evaluations of landscape stimuli and the degree of fixation coincidence when viewing the spaces, among which the rate of distant clarity and the degree of fixation behavior coincidence were significantly and positively correlated. Meanwhile, at the second viewing, the number of favorite elements in the lookout space, which belongs to high-preference spaces, noticeably increased.

## 1. Introduction

With the rapid development of material culture, people’s demand for forests has changed from the timber economy of the past [1,2] to the deep exploration of sustainable utilization, such as recreation, tourism and health care [3,4,5,6,7,8,9]. In addition, due to the globalization and normalization of the COVID-19 epidemic, many countries and regions have taken control measures against the spread of the epidemic. People’s pursuit of their own safety and health [10] makes them more inclined to cancel their long-distance travel plans and choose forest landscapes with rich natural resources close to their own residence [11]. This also means that it is of practical significance and necessity for managers and designers to explore the visual behavior characteristics of people’s repeated viewings of suburban forest landscape resources and know what kind of landscape can attract people for repeated viewings, which will help promote the sustainable utilization of suburban forest resources.

However, as pointed out by Barnes et al., the basic question of “what and how people see” in forest landscapes has not been answered using scientific quantitative methods [12]. From the perspective of forest resource management and evaluation, landscape architects can obtain public feedback on landscapes by understanding people’s perceptions and observation modes to better carry out follow-up planning and management for landscapes. In this process, eye tracking technology is a useful tool to detect people’s observations [13], and cross-recurrence analysis is a scientific method to detect the degree of fixation behavior coincidence when people view the landscape many times [14].

Currently, eye tracking technology is widely used in landscape architecture. In the 1990s, this technology was first applied to forest landscape research by European scholars De Lucio et al. and Hughes et al. [15,16]; thereafter, the application of eye movement technology in landscape fields became increasingly abundant and diverse. Nordh et al. used eye movement technology to explore the restoration of urban pocket parks and pointed out that the area of ground cover plants was positively correlated with restoration [17].

In terms of population attributes, research by DuPont et al. and Zhang et al. showed that the visual behaviors exhibited by groups vary with the backgrounds. Experts with landscape backgrounds are more inclined to explore the entire scene on a large scale, hence people’s psychological evaluations of forest landscapes will contain a “professional effect” [13,18].

In the field of visual behavior characteristics, Amati et al. found that there were individual differences among people, as well as differences in their preferences for skies, shrubs and stairs [19]. In addition, Gao et al. and Zhou et al. analyzed the relationships among the types of forest landscape spaces, the characteristics of their elements and people’s visual behaviors and preferences. The results showed that the characteristics of landscape elements significantly affect people’s visual behaviors and cognitive preferences, and people spend more time with elements of high complexity and proportion [20,21].

As for seasonal changes and regional landscape features, Paraskevopoulou et al. found that deciduous trees played a positive role in rehabilitation therapy [22]. Zhang et al. found that designs with regional features had positive significance for urban landscape protection and reconstruction [23]. Millar et al. found that relatively undeveloped and agricultural landscapes attract the dynamic attention of participants [24].

In addition, Liu et al. combined vision with audition to explore the influence of sound on visual behavior for various landscapes and found that light music and the chirping of insects and birds make people’s visual behaviors more relaxed [25]. Williams and Castelhano explored the different eye movement challenges of dynamic landscapes and static landscapes and proposed the synchronization challenge of eye movement when viewing dynamic landscapes [26].

Comparing nature and buildings, Van den Berg et al. found that natural scenes are more likely to attract people’s visual attention than built scenes, and the psychological and emotional restorative effect is improved [27]. Furthermore, people’s visual behaviors when viewing natural landscapes with high restorative ratings are significantly different from those when viewing urban scenes with low restorative ratings. When viewing natural scenes, people spend less time viewing, and the viewing takes less effort [28]. 

These studies indicate that eye movement technology is a relatively mature technology in the field of landscape architecture, and multidimensional research has already been carried out. However, throughout these studies, we find that researchers tend to study people’s single viewing behaviors in scenes, but the visual behavior of people’s repeated viewings in scenes needs to be further explored.

In the 1980s, David Noton predicted that people would have similar eye movement behaviors when viewing the same object multiple times [29]. Subsequently, researchers began to discuss the visual behavior of repeated viewings for portraits, art, nature and industry [30,31]. These studies explored the differences in visual behavior characteristics based on the similarity of scan paths and pointed out that, although participants had different eye movement behavior patterns when viewing different objects, the visual behavior patterns of those who viewed the same object many times were similar.

For example, Humphrey and Underwood, David Noton, Zeni and Hannah Faye Chua et al. studied from different perspectives aimed at exploring people’s visual behavior characteristics. They constructed related schemas on repeated viewings and searched for the reasons why people’s visual behaviors are alike, i.e., they pay more attention when repeatedly viewing, and how they distribute their attention [29,31,32,33]. Most importantly, these studies pointed out that there are indeed interesting differences in visual behaviors when viewing repeatedly. From the psychological point of view, these differences are partly due to the increase in familiarity caused by repeated viewing. After people have deeper understandings of the objects, their viewing mode will change, i.e., people will gradually shift from the bottom-up exploration mode guided by the salience of the object to the top-down observation mode with more subjective preferences. This means it is necessary to further understand the psychological evaluations of people on the basis of exploring the changes in visual behavioral patterns when exploring the repeated viewing of forest landscapes, so as to attempt to explore the reasons for the visual changes.

Generally, in research on repeated viewings, most of people’s visual behaviors are correlated in repeated viewings, but the kind of correlation is a topic worthy of further discussion. In this process, the cross-recurrence analysis in recursive quantitative analysis is the key means to analyze the coincidence degree of visual behaviors in repeated viewings.

As an analytical method for studying the quantification of dynamic systems, the recurrence quantification technique was first applied in physics and physiology and was used to explore behavior that changed discernibly throughout a time series [34,35,36]. Since Eckmann first proposed this method in 1987, it has become a mature system and has gradually been applied to various fields for more than 30 years [34].

In 2013, Anderson et al. applied recurrence quantification analysis in the field of eye movement to explore the overlap of individuals’ gaze sequences and put forward the cross-recurrence analysis in eye movement to compare the similarities of different individuals’ gaze sequences. In confirming its usability, the free viewing mode and limited viewing mode were combined; this revealed that people exhibited more backsight behaviors when viewing freely and that the overlap of visual behaviors was significantly related to the attributes of stimulation [14].

The recurrence quantification technique uses the fixation distance method, which addresses the inability of the traditional basic index system in eye movement, to compare the fixation trajectory between objectives and solves the deviation of the definition of “adjacent” in repeated viewing research [14].

Through the review and summary of the above research, we find the following:(1)Many researchers have explored the relationship between people’s visual behaviors in a single viewing of landscapes and evaluated the psychological associated factors;(2)Research on repeated viewings has not systematically examined landscapes. In today’s normal epidemic situation, people are bound to visit the same forest landscape repeatedly. In this context, the questions of “What visual behaviors do people exhibit when they view a landscape many times?” and “What is the relationship between them?” are important. Other issues still need to be discussed in depth;(3)Because people’s preferences for landscapes are related to their sightseeing intentions and visual behaviors in a single viewing [21,37], there are also differences in people’s visual behaviors across landscapes during repeated viewing experiments [31]; however, the relationships among people’s visual behaviors, psychological evaluations and preferences in repeated viewings is still unclear.

Therefore, from the perspective of landscape preference, this study explores the changes in individuals’ visual behavior characteristics and psychological characteristics under different preferences when viewing forest landscapes during repeated viewings. It attempts to clarify the kinds of landscapes people prefer or view repeatedly at time intervals. The study also explores the kinds of visual behaviors exhibited during viewing to explore how these behaviors and evaluations influence the planning and design trend of forest landscapes.

According to the relationship between preference and visual behavior summarized by other authors, we posit the following hypotheses (Figure 1):

**Hypothesis** **1** **(H1).**
*There is a significant difference in the similarity of visual behavior characteristics of forest landscapes after repeated viewings for high- and low-preference landscapes, and there is a positive correlation between similarity and preference.*


**Hypothesis** **2** **(H2).**
*There is a significant difference in the changes in individuals’ psychological evaluations of forest landscapes after repeated viewing between high- and low-preference landscapes, and there is a positive correlation between similarity and preference.*


**Hypothesis** **3** **(H3).**
*Individuals’ psychological evaluations of different forest landscapes lead to different visual behaviors.*


## 2. Materials and Methods

### 2.1. Study Area

To make the experimental samples universal and representative, this study selected five famous forest parks in eastern Liaoning Province as study areas: Shenyang National Forest Park, Heyi National Forest Park, Greenstone Valley National Forest Park, “Liberation Forest” in Caohekou and Phoenix Mountain National Scenic Area (Figure 2). These forests have the vast majority of forest landscape features. At the same time, they are all located in the suburbs of cities, have convenient traffic conditions, are within 60–100 km from the city center and thus represent forest landscape resources in the suburbs of the city.

### 2.2. Stimulus

Many studies have shown that green plants have good rehabilitative effects [38], and they are more favored by people than other colored plants [39,40,41]. Some studies have explored people’s evaluations of plants in different seasons and found that the ratings of plants in spring, summer and autumn are similar and higher than those in winter [42]. In addition, due to the age limit of the participants, in this study, we explore young people’s visual behaviors in spring when green plants grow.

A large number of studies have explored and confirmed the relationship between on-site surveys and photos and found that photos can replace on-site surveys to some extent. Therefore, in our research, photos were chosen to be the experimental stimuli [37,43,44,45]. The landscape photos were taken on sunny and cloudy mornings in the spring of 2021 (April-June) [46,47]. We selected 24 sample plots in these parks and took 56 photos in total. Then, through an online questionnaire, we conducted the first-stage preference survey of landscape architecture majors and nonmajors and obtained 90 valid questionnaires. Subsequently, three photos given high- and low-preference ratings in the questionnaires were selected as the experimental stimuli, yielding six photos in total (Figure 3).

### 2.3. Participants

It has been proven that the survey results of college students agree with young people’s actual aesthetic preferences and visual behaviors [20,21,37]. The data of 52 participants were valid (including 24 males and 28 females; 24 majoring in landscape architecture and 28 other majors; 36 undergraduate students and 16 graduate students). All the participants are Chinese people.

### 2.4. Experimental Design

This study includes two parts: an eye movement experiment and a psychological evaluation.

Before the experiment, the participants were simply told the purpose of the experiment and informed that the experiment should be repeated at the same time the following week. The eye movement experiment of each experiment was carried out in a special and fixed eye movement laboratory. After the eye movement experiment was finished, the experimental pictures were randomly shown again, and the evaluation questionnaire was completed. The participants were given gifts after they completed the experiment. The specific experimental process is shown in Figure 4. 

### 2.5. Selection of Indexes

#### 2.5.1. Selection of Eye Movement Indexes

To explore the correlation of the visual behavior characteristics of the participants when they repeatedly viewed forest landscapes, this study utilizes the cross-recurrence analysis index system (CRA). This system processes the fixation point to directly compare the similarity of fixation behaviors in repeated viewings [14,48]. In this study, four indexes (REC, DET, LAM and CORM) were selected, and their algorithms and meanings are shown in Table 1.

In our research, the same participants’ fixation sequences on the first and second viewings were selected to analyze the coincidence degree.

#### 2.5.2. Selection of Psychological Cognition Evaluation Indexes

The questionnaire consists of three parts. The first part surveys demographic information; the second part is the landscape evaluation using a Likert scale (scoring from low to high is 1–7); and the third part contains multiple-choice questions (see Table 2 for details).

The landscape evaluation indexes in this study were extracted from the previous landscape evaluation system. The participants were asked to evaluate and score the photos in terms of four aspects: spatial characteristics, color, landscape changes and overall evaluation [28,49].

#### 2.5.3. Definition of Terms

In this study, subjective evaluation includes two terms: one is the psychological evaluation, and the other is the cognitive evaluation.

Psychological evaluation: It includes two parts, namely the cognitive evaluations of landscapes (including evaluations about space, color and landscape change overall in the form of a Likert scale with scores from 1–7) and the recordings and descriptions of people’s preferences for landscape elements (in the form of multiple-choice questions). It reflects the evaluations of landscape characteristics and the preference for elements.

Cognitive evaluation: It refers to the cognitive evaluations of landscapes (including evaluations about space, color and landscape change overall in the form of a Likert scale with scores from 1–7), which only reflects the evaluation of landscape characteristics.

### 2.6. Analysis and Statistics

In order to evaluate Hypothesis 1 (H1), according to the characteristics and attributes of the data, a nonparametric K-W test is used to compare the differences in the visual behavioral coincidence degree of all participants when repeatedly viewing. 

In order to answer Hypothesis 2 (H2), a descriptive statistical analysis was used to compare all the landscape elements that the participants liked and disliked when they repeatedly viewed the landscapes. This method was also used to analyze their changing trends with the number of times they viewed them; a nonparametric Wilcoxon Signed-Rank test was used to explore the changes in psychological evaluations on the first and second viewings.

In order to answer Hypothesis 3 (H3), a Spearman correlation analysis was used to explore the correlation between the subjective psychological evaluations and objective visual behaviors of the participants when viewing landscape stimuli.

## 3. Results

In this study, 59 undergraduates and postgraduates from Shenyang Agricultural University took part in the experiment. Those who did not participate in the second experiment (3), those whose eye movement results could not be derived (3) and those whose data were invalid (1) were screened out. In the end, the data from 52 participants in total were valid.

### 3.1. Analysis of the Characteristics of Visual Behavior When Repeatedly Viewing Forest Landscapes

To begin, we tested whether the data obey the assumption of normal distribution and homogeneity of variance and found that all the data do not obey normal distribution, *p* < 0.050; we used the K-W test to analyze the visual behavior coincidence degree. The result is shown in Table 3. From the results we can see the following trends:

The recurrence (REC) of the dynamic waterscape HP1 (17.125) with high preference was higher than that of HP3 (13.045) and LP1 (11.495), *p* < 0.050, than that of LP2 (11.910) and that of LP3 (9.615), *p* < 0.001. In addition, the static waterscape HP2 (15.405) with high preference was higher than all landscapes with low preference for LP1 (11.495), LP2 (11.910) and LP3 (9.615); HP3 (13.045) was higher than that of LP3 (9.615). There was no significant difference among the other landscapes (*p* > 0.050). The recurrence of all six stimuli in repeated viewings was between 10% and 20%, which means that the coincidence degrees of the fixation points in repeated viewings were generally low, and the participants were more inclined to view the areas they had not seen in the first viewing. The determinism (DET) of the dynamic waterscape HP1 (50.230) was higher than that of the lookout landscape HP3 (38.035) and of all landscapes with low preferences for LP1 (38.160), LP2 (32.320), *p* < 0.050 and LP3 (30.610), *p* < 0.001. On the other hand, the determinism of the static waterscape HP2 (42.230) was significantly higher than that of LP2 (32.320), *p* < 0.050 and LP3 (30.610), *p* < 0.001. The determinism of HP3 (38.035) and LP1 (38.160) were higher than that of LP3 (30.610), *p* < 0.050. There was no significant difference among the other landscapes (*p* > 0.050). The laminarity (LAM) of the dynamic waterscape HP1 (66.670) was significantly higher than that of the lookout landscape HP3 (50.610) and of all landscapes with low preferences, i.e., LP1 (50.000), *p* < 0.050, LP2 (45.380) and LP3 (43.590), *p* < 0.001. On the other hand, the laminarity of the static waterscape HP2 (57.935) was significantly higher than that of LP2 (47.102) and LP3 (40.748). In addition, LP3 (40.748) was significantly lower than all those with high preferences, and it was also significantly lower than LP1 (50.000). The mean values and medians of DET and LAM were both in the range of 30%~70%, which means that in repeated viewings, the participants exhibited similar visual behaviors when viewing the same fixation points.

Finally, Figure 5 shows that the start of the first viewing of the dynamic waterscape was sooner than the second viewing (μ = 0.515). The static waterscape (μ ≈ 0) can be regarded as having no precedence relation in the gaze sequence. The lookout landscape (μ = −1.556) and all the low-preference landscapes (μLP1 = −1.901; μLP2 = −1.869; μLP3 = −0.964) indicated that the start of the second viewing was sooner than that of the first viewing. This means that only the visual behavior in the waterscape space was more sensitive for the first viewing, which started relatively late for the second viewing. However, for the stimulation of the lookout landscape and all low-preference landscapes, the visual behavior of the second viewing was more sensitive, and the participants started searching the landscape more quickly.

This means that the participants were more likely to exhibit similar visual behaviors when they repeatedly viewed the waterscapes one week later, but they started viewing sooner in the first viewing. The participants show less similarity in visual behaviors for the lookout landscape and even less for the broadleaf forest landscapes, but they were more sensitive to starting at the second viewing for these landscapes. However, there were significant differences in the broadleaf forest landscapes, which needs to be discussed from other perspectives. We used ImageJ v1.8.0.112 to process the three broadleaf forest stimuli and compare the colors of the three stimuli through the color histogram, as shown in Figure 6.

The histogram shows that the color tendencies of the broadleaf forests LP1 and LP2 were significantly different from those of the broadleaf forest LP3: the green value of LP3 was significantly lower than that of LP1 and LP2 at each brightness level. Above all, we found that the coincidence degree of the participants’ visual behaviors in the high-preference landscapes were greater than or equal to those in the low-preference landscapes.

We also compared the visual heatmaps of the participants when they viewed the landscapes for the first time and for the second time (Figure 7). It can be seen that the participants’ visual concerns are relatively concentrated in the center of spaces or the vanishing point of sight in spaces during the first viewing. However, at the second viewing, the participants’ visual concerns in the same space showed a tendency towards diffusion. 

This phenomenon is reflected in the static water space HP2, the overlooking space HP3 and the broad-leaved forest LP2 with a strong sense of openness. The participants’ visual attentions are more inclined to the mountains (LP2, LP3), the sky (LP3) and the tree crowns (LP2, LP3) in the spaces at the second viewing. That is to say, at the second viewing, subjects’ visual attention to landscape spaces shows a trend “from partial appreciation to overall appreciation”. 

In summary, there is a significant difference in the similarity of visual behavior characteristics of forest landscapes after repeated viewings for high- and low-preference landscapes; therefore, Hypothesis 1 holds.

### 3.2. Analysis of the Characteristics of Psychological Evaluation When Repeatedly Viewing Forest Landscapes

#### 3.2.1. Analysis of the Characteristics and Differences in Overall Cognitive Evaluation When Repeatedly Viewing Landscapes

We first tested whether the data obey the assumption of normal distribution and homogeneity of variance and found that all the data do not obey normal distribution, *p* < 0.050. Hence, we used the Wilcoxon test to compare the cognitive evaluations for repeated viewings (see Table A1 for details). Except for the color evaluations and plant diversity, all the landscape evaluation indexes of the participants showed no significant changes in repeated viewings. Only the color evaluation of HP2 and LP2, and the plant diversity of HP1 decreased significantly at the second viewing. The conclusion was drawn that repeated viewing with a one week interval does not significantly affect young people’s cognitive evaluations of forest landscapes.

On this basis, a group of boxplots was drawn to further analyze the differences in cognitive evaluations between the six landscapes in the first viewing; the differences in those in the second viewing; and to explore the more subtle features of the six landscape stimuli that can explain visual behaviors. The results are shown in Figure 8.

In terms of plant diversity, LP3 (4.192) was lower than LP1 (4.308) and LP2 (4.327); for color richness, LP3 (4.981) was lower than LP1 (5.096) and LP2 (5.327); and with respect to spatial hierarchy, LP3 (3.788) was lower than LP1 (4.058) and LP2 (3.942), but there were no significant differences (*p* > 0.050).

On the whole, similar to the results obtained in previous work, the reviewing intention for landscapes with high preference was significantly higher than that for landscapes with low preference, and the dynamic waterscape HP1 (5.846) in landscapes with high preference was also significantly higher than that in the lookout landscape HP3 (5.221), *p* < 0.050. This shows that there was grading in the landscapes with high preference.

In addition, in terms of plant diversity and spatial hierarchy, lookout landscape HP3 was rated at the same level as all low-preference landscapes.

The spatial openness of the dynamic waterscape HP1 (5.308) was lower than that of the static waterscape HP2 (6.365) and the lookout space HP3 (6.538). The distant clarity of HP1 (5.308) was lower than that of HP2 (6.096) and HP3 (5.654), but there was no significant difference (*p* > 0.050).

#### 3.2.2. Analysis of the Characteristics and Differences in Landscape Elements Participants Focused on When Repeatedly Viewing Landscapes

Descriptive statistics are given for the attention to landscape elements in repeated viewings of different landscape stimuli in Figure 9. The figure shows that at the second viewing, the number of preferred elements of the waterscape (HP1 and HP2) showed a constant or decreasing trend, while the number of disliked elements showed a constant or increasing trend. However, it is interesting to note that the change trend of the likes and dislikes of HP3 was opposite to that of the former. The number of likes and dislikes increased noticeably (from 60 to 117), mainly for mountains and distant elements. Although the total number increased significantly, the value was still temporarily smaller than that of waterscapes.

At the second viewing in the low-preference stimuli, the number of likes for broadleaf forest LP3 decreases considerably (from 86 to 58), which is mainly reflected in the plant elements in the scene. However, LP1 and LP2 in the broadleaf forests did not change considerably. These results are consistent with those shown in Figure 7.

Above all, there is not a significant difference in the changes in individuals’ cognitive evaluations of forest landscapes after repeated viewing between high- and low-preference landscapes, but the likes and dislikes of landscape elements changed at the second viewing. Therefore, Hypothesis 2 does not hold.

### 3.3. Analysis of the Relationship between Visual Behavior Characteristics and Psychological Evaluation Characteristics When Repeatedly Viewing Different Forest Landscapes

To explore the relationships among REC, DET, LAM, CORM and cognitive evaluations of the landscape stimuli, correlation analysis of the overall six landscapes between CRA indexes and cognitive evaluations was established and is shown in Figure 10.

Although the Wilcoxon Signed-Ranks test of the two cognitive evaluations showed no significant difference, we further analyzed the results of the two evaluations and the characteristics of visual behaviors. We found that there are indeed differences. The correlation between cognitive evaluations and visual behaviors under the first viewing was relatively large and more significant, and many indexes of visual behavior coincidence (REC, DET, LAM) were all related to the indexes of cognitive evaluations. However, at the second viewing, the correlation coefficient tended to weaken. At the same time, compared with the cognitive evaluations at the first viewing, only REC, the index of fixation point coincidence, was related to cognitive evaluation indexes at the second evaluations.

On the other hand, the correlation between the entire evaluations and the eye movement indexes is positive, which means that the higher young people’s evaluations of the landscapes are, the higher the degree of fixation coincidence in repeated viewings, and the more inclined to view the areas they had seen last time. Moreover, the feature evaluations were only related to REC: When young people evaluated the landscape features, it was only related to the coincidence degree of fixation points, but the spatial evaluations and overall evaluations of the landscape were mostly related to the degree of various visual behavior coincidence. This means that when evaluating these dimensions, young people would be influenced by regularly continuous visual behaviors.

In addition, we found that whether it was the first evaluation or the second, the distant clarity index is affected by the coincidence of all three visual behaviors. In other words, with the improvement of young people’s awareness and familiarity with spaces, this landscape space index represented by the visual clarity (SDL) in landscape spaces had always significantly influenced young people’s continuous visual behaviors.

Combining the above two points, we think that after having a certain familiarity with the landscapes, young people’s visual behavior sequence was disrupted and reorganized, and young people tended to shift from the initial local exploration strategy to the macro-global exploration strategy. This had also been verified in Figure 7.

Considering the relationship and logic between viewing times and visual behaviors, we guess that the cognitive evaluations of the first viewing can play a certain guiding or predicting role in the subsequent repeated visual behaviors. This may be the reason for the difference between the correlation of two evaluations and the fixation coincidence degrees. According to the correlation, the fixation coincidence degree of the second and the third viewings can be further inferred from the evaluations of the second viewing.

Above all, individuals’ psychological evaluations of different forest landscapes correlated with visual behaviors; therefore, Hypothesis 3 holds.

## 4. Discussion

### 4.1. The Difference in Visual Behavior Characteristics in Repeated Viewings across Landscape Stimuli

Our results showed that the overall coincidence degrees in repeated viewings were lower; among them, dynamic waterscape ≈ static waterscape > lookout space ≈ broadleaf forest 1 ≈ broadleaf forest 2 > broadleaf forest 3 (Table 3).

In the results, the visual behavioral coincidence degrees of the landscapes with high preference were not lower than those of the landscapes with low preference. However, we think the result is more complicated than we predicted. Combined with the values of CRA, the lookout landscape can be regarded as the outlier in the landscapes with high preference, while the broadleaf forest LP3 can be regarded as the outlier in the landscapes with low preference. We speculate that the reasons are as follows:The higher the preference for a landscape, the higher the coincidence degree of young people’s visual behaviors, which produces a similar repeated regression for viewing the landscape. However, when compared with waterscapes, lookout landscapes have lower overall landscape richness, insufficient details and relatively less attractive content. Therefore, the regression behaviors decrease significantly at the second viewing. Liu et al. found that when people view lookout landscapes, when compared with other landscapes, they tend to feel relaxed and peaceful, and their psychological burden is smaller. Especially after becoming familiar with a landscape, a relaxed state of mind is generated more quickly, so the visual regression behavior is more reduced [25]. We guess that because mountaintop landscapes can be reached only after a long journey over a rugged mountain road, people are more inclined to exhibit relatively relaxed visual behavior when viewing them; the effect is similar to the theory of “suppress first to promote” in the field of psychology [50]. In addition, when individuals view waterscapes, they are attracted by their rich details and elements. There is also related theoretical support in the field of psychology: within a certain range that does not exceed the threshold, the complexity of landscapes is positively related to their attractiveness [51];All three landscape stimuli with low preference were broadleaf forests, but the plant color (RGB) of broadleaf forest three was different from that of broadleaf forest one and two, and the pixel value of green in LP3 was relatively small; however, there was no obvious difference among them. Therefore, we think that a relatively high saturation of green attracts people more in the same kind of landscape, which is consistent with the conclusions of Neale et al., Serra et al. and Wang et al. [52,53,54]: Green plants are more attractive than plants with other colors. People are attracted when they view green plants, resulting in more positive feedback;Although there were significant differences in the coincidence degree of repeated viewings of the six landscape stimuli, they were generally low, which is somewhat consistent with Noton’s conclusion in 1970: Similar visual behaviors occur, but mainly in the initial learning stage [29]. Our findings show that there was a significant difference in the coincidence degree of repeated viewings between landscapes under 15 s gaze conditions. Menon and Levitin et al. pointed out that previous experience allows participants to form expectations or assumptions about objects. This expectation or assumption restricts the cognitive level of participants to objects. According to this theory, the participants had a “Peak shift” when they viewed the landscapes for the second time. This means that when they viewed the picture for the second time, based on the experience of the first viewing, they had formed their own inherent understandings of the spaces to a certain extent, which in turn led to similar visual behaviors in the space at the second viewing. Because perceptual information is caused by the interaction between realistic stimulus information and memory information, and people’s visual behavior is based on their perceptual information, the visual behavior at the second viewing is similar to that at the first viewing [55]. In addition, we suspect that the overall low degree of coincidence may be due to people keeping target information in visual short-term memory (VSTM), which is relatively persistent but has a limited capacity when scanning [56]. This may mean that the established memories of people may be replaced by new memories in a certain time interval, resulting in the gaze mode in memory not being followed subconsciously.

### 4.2. The Differences in Psychological Evaluations in Repeated Viewings across Landscape Stimuli

At the second viewing, the cognitive overall evaluations in repeated viewings did not change, but the characteristics of landscape elements the participants paid attention to changed.

#### 4.2.1. Characteristics and Differences in Cognitive Overall Evaluation When Repeatedly Viewing Landscapes

For the cognitive evaluation part of the Likert scale, the experimental results were quite different from expectations: with an increase in familiarity, the participants’ feature evaluations and comprehensive evaluations of landscapes did not change significantly. There are several possible reasons:For a given landscape, viewing it again after one week is not enough for young people to have a high degree of familiarity which affects their comprehensive senses, thus causing negative emotions such as aversion and conflict in response to the landscape. We hypothesize that this may be because people tend to remember the content they like and ignore the content they do not like or find boring. In addition, after analyzing the characteristics of three low-preference landscapes, the three broadleaf forest landscapes had high similarity. According to the eclipsing effect in psychology, when the similar contents are concentrated, it is easy for individuals to mix them in their memories, so they are difficult to regenerate and more likely to cause forgetfulness. A review by Henry L also mentioned this possibility, which may also be one of the factors affecting the participants’ memories [57];R. Aiken pointed out that a stable questionnaire will not reveal large differences when the time interval of repeated experiments is relatively short, which is also reflected in our research; however, there are some interesting subtle changes in our analysis [58].

Although the cognitive evaluation changes were not significant, we analyzed the subtle changes and possible trends and found that the spatial indexes (spatial openness, distance clarity, spatial neatness and three-dimensional ratings) and color indexes (color richness, color brightness) of the lookout landscape were slightly increased with reviewing; this result is rather interesting. According to Appleton’s “Prospect–Refuge” theory, people prefer places where they can “watch but not be seen” [59]. Compared with all other landscape stimuli, the lookout space had a higher viewing angle that allows the distant horizon to be viewed and is located at a high position, which is difficult for others to find at low altitudes. In spring, when the temperature is moderate, this kind of landscape increasingly attracts young people to repeatedly visit, and their evaluations are continuously improved.

In addition, the evaluation of the broadleaf forest LP3 was always the lowest, and the decreasing trend was the largest. According to the prospect–refuge theory, a dense canopy can provide a good shelter environment [59]; however, the trees were too lush to provide a good lookout angle, so the overall evaluations of the three broadleaf forests were low. In addition, according to Gestalt theory, people’s eyes tend to see close or complete figures [60]. Compared with the broadleaf forests one and two, the overall layout and morphological composition of the broadleaf forest three did not allow a complete figure to be formed, which may be the psychological change caused by this phenomenon.

#### 4.2.2. Characteristics and Differences in Landscape Elements Focused on When Repeatedly Viewing Landscapes

The number of likes for the high-preference landscapes was significantly higher than that for the low-preference landscapes, while the number of dislikes was significantly lower than that for the low-preference landscapes. At the same time, the overall elements of likes at the second viewing of the lookout landscape increased significantly, and the mountains and the distant scenery elements increased the most.

The research of Liu et al. affirmed the attraction of lookout spaces for single viewings [25], and the research team of Tingting Mou also drew relevant conclusions for mountain landscapes [61]. In our research, when viewing the lookout landscape for the second time, the favorite landscape elements greatly increased, indicating that the participants’ positive evaluations of the lookout landscape increased significantly with the increase in viewing times; young people may also be attracted to view such landscapes many times (Figure 9A). Zhang et al. pointed out that open space provides an inclusive visual experience, while depth of field enriches content and visual perception. This further explains Appleton’s prospect–refuge theory [62]. In our research on the stimulation of natural forest landscapes, the lookout landscape also presents a similar effect, which is the same as the previous conclusion.

This shows that lookout landscapes are highly attractive for repeated viewings, and adding clearly visible distant scenery and mountains to other landscape types may also result in better evaluations. Although the total number of likes for the lookout space increased significantly, its value was still smaller than that of the dynamic waterscape and static waterscape; this shows that the attractiveness of the lookout landscape was still smaller than that of waterscapes at the second viewing, but its growth trend was positive, and it is expected to surpass the latter after additional viewings.

In conclusion, lookout landscapes may have a more positive trend of attracting young people to repeated visits.

At the second viewing, the number of likes for the broadleaf forest LP3 dropped sharply and became significantly lower than those for the broadleaf forest LP1 and the broadleaf forest LP2. The main decreased elements in LP3 were shrubs and ground cover plants, while in the other two, there was little change in those elements. Combined with Table 3 and Figure 8, we speculate that the shape and composition of shrubs and ground cover plants in broadleaf forests greatly affect young people’s cognitive evaluations, the similarity of fixation points and fixation behaviors in repeated viewings. This also confirms the research conclusions of Li and Lv that the geometric relationships of landscapes affect people’s sensory preferences [63,64].

### 4.3. Interaction between Visual Behavior and Psychological Evaluation

From the correlation results, all the indexes of psychological evaluation were significantly correlated with REC but not with DET and LAM (Figure 10).

This shows that the higher the cognitive evaluation is, the higher the coincidence degree of fixation points; that is, the more attractive a landscape is, the more obvious the areas of interest that attract young people’s attention, and the more attention they allocate to these areas when they are viewed repeatedly. However, landscapes with low cognitive evaluation were viewed by young people more randomly when viewed repeatedly due to their lack of an attractive area of interest. Regardless of the cognitive evaluation, when young people repeatedly view these landscapes at certain time intervals, they exhibited different visual behaviors. Williams D.R. put forward the theory of place attachment in 1989. The theory holds that place attachment can increase young people’s sense of belonging and identity with the scene; in other words, users’ higher familiarity with places means that they have better cognitive evaluations [65]. In this case, they also allocate more attention to these places. In addition, when individuals have positive emotions, they will adopt more comprehensive processing strategies, while when individuals have negative emotions, they will adopt more specific and detailed processing strategies [66]. This also leads to significant differences in the visual behaviors of landscape spaces under different preference evaluations.

In other words, the regression behavior of eye movement is related to young people’s cognitive evaluations and preferences for landscape elements, which is consistent with the potential relationship between eye movement behavior and psychological evaluation proposed by previous studies [18,20,21,37]. Figure 11 shows the relationship between participants’ visual behaviors and landscape element preferences for repeated viewings. We think that eye movement influences cognitive evaluation and elemental likes and dislikes at the first viewing and promotes eye movement at the second viewing. The cognitive evaluation at the first viewing basically determines the cognitive evaluation and elemental likes and dislikes at the second viewing. By comparing the first and second eye movements and evaluations, the results reflect the participants’ intentions to view the landscape again.

### 4.4. Limitations

From the perspective of landscape stimulation selection, the selections of high-preference landscapes was rich and reasonable, but there was no significant difference among the three landscapes with low preference; therefore, it is difficult to compare and study them. In addition, the color changes caused by seasons affect people’s visual behaviors, and we will discuss the differences further in the future. The differences between on-site surveys and photos will also have a certain impact on the experimental results. However, finance and data acquisition are inevitable problems. This would be our future research topic;On the selection of participants, this study mainly explored the visual behaviors and psychological evaluations of young people under repeated viewings. In future research, the scope could be further expanded to people of all ages;Regarding the experimental design, this study explored only the similarities and differences in visual behaviors between the first viewing of the forest landscape and the second viewing one week later. Future research should further explore the similarities and differences in visual behaviors with increased viewings, as well as those of visual behavior at various time intervals;Regarding the research method, the CRA method used in this study explored the similarity of visual behaviors, but it could not analyze how the parts differ. It is possible that there is a more scientific and comprehensive way to explore this aspect in the future.

## 5. Conclusions and Suggestions

### 5.1. Conclusions

In this study, cross-quantification analysis was used to explore whether repeated viewings of forest landscapes affect young people’s visual behaviors and cognitive evaluations. The study aims to provide suggestions on return visits for the transformation of forest landscape resources and to promote their sustainable value.

The evidence provided by our study shows that specific features in the landscapes can be translated into specific visual behaviors, and this can influence the decision to return to the same place:With the increase in viewing times, all kinds of regression behaviors show a decreasing trend. Young people are more inclined to view areas that they have not viewed before. At the same time, during a second viewing, when young people view an area they did not see in the first viewing, the coincidence degree of visual behavior is generally low, and there are obvious differences across landscapes;With increased viewing times, young people’s cognition evaluations of landscapes did not significantly change. However, in the process of repeated viewings, young people are more inclined to view waterscapes and lookout landscapes, and lookout landscapes have a more positive tendency to increase individuals’ preferences during repeated viewings;The regression behavior of eye movement is related to young people’s cognitive evaluations and preferences for landscape elements. There is a significant positive correlation between cognitive evaluation and the coincidence degree of fixation points when viewing landscapes, but only distant clarity and the coincidence degree of all kinds of fixation behaviors are significantly and positively correlated. In addition, the landscape type is also related to the likes for specific landscape elements: with increased viewing times, only the number of likes for the lookout landscape increases noticeably.

### 5.2. Suggestions

We suggest that in the process of forest landscape planning, waterscape spaces and lookout spaces be placed at nodes with high accessibility in forests, such as near the entrance or exit of forests. Auxiliary facilities should be provided to reach such spaces.

In addition, different spaces should be considered separately (Figure 12):When designing dynamic waterscapes and lookout spaces, more colorful trees should be planted to improve the color richness and layering of the scenery. At the same time, scenic spots with more rolling mountains should be selected to improve the spatial hierarchy, thus enhancing tourists’ revisit intentions;When designing static waterscape spaces, it is necessary to plant more brightly colored plants, increase the types of vegetation and maintain a high degree of openness to enhance tourists’ revisit intentions;When designing the undergrowth landscape space represented by broadleaf forests, more consideration should be given to the composition and proportion of shrubs and ground cover plants. Vegetation with higher green pixel values should be selected, and plants with different shapes should be matched to increase the complexity of the space to improve the attraction of the landscape.

## Figures and Tables

**Figure 1 ijerph-20-04753-f001:**
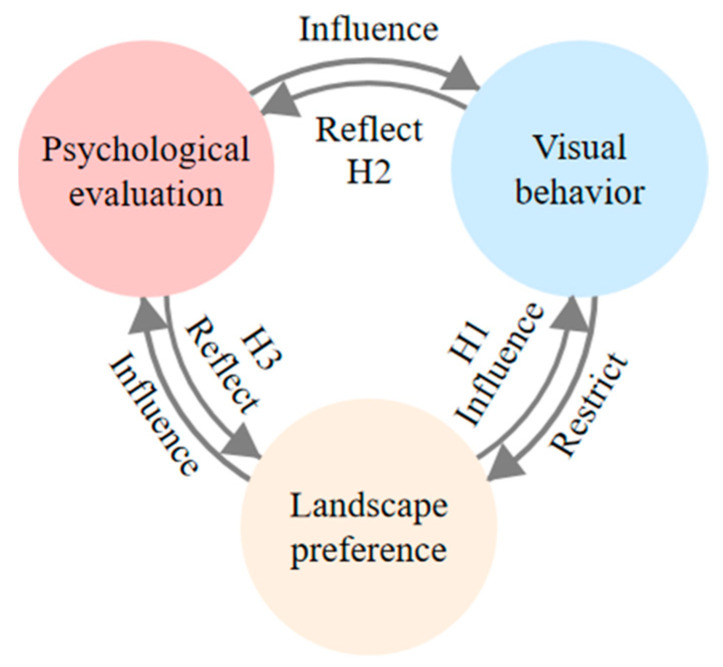
Research hypothesis diagram. This figure was created by the author.

**Figure 2 ijerph-20-04753-f002:**
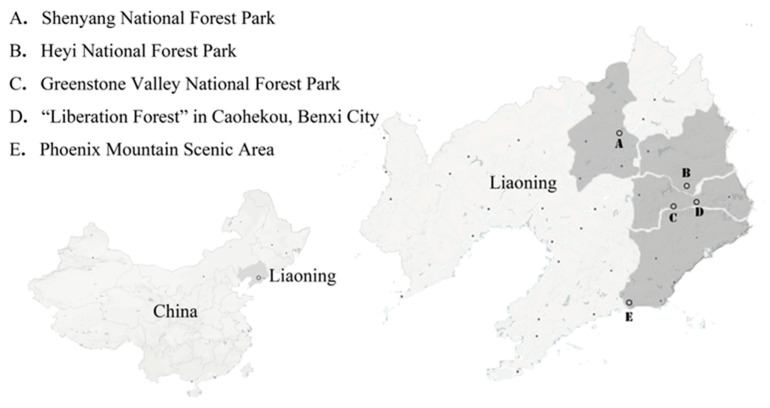
Locations of the study sites. A. Shenyang National Forest Park (covering an area of approximately 9.33 square kilometers and the main stand is black pine forest). B. Heyi National Forest Park (covering an area of approximately 18.68 square kilometers with natural secondary forests and plantations). C. Green Stone Valley National Forest Park (covering an area of approximately 20 square kilometers; it is a typical mountain-type forest). D. “Liberation Forest” in Caohekou (covering an area of approximately 0.22 square kilometers; it is an artificially planted Korean pine forest). E. Phoenix Mountain National Scenic Area (covering an area of approximately 216 square kilometers, it has abundant and famous waterscapes). This figure was created by the author.

**Figure 3 ijerph-20-04753-f003:**
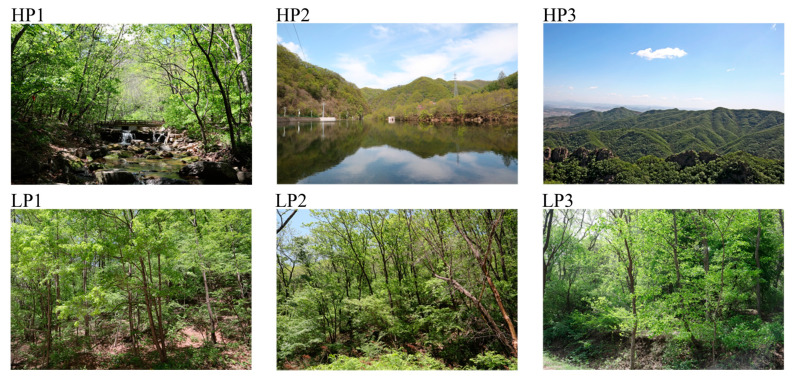
Experimental stimuli. Three high-preference landscapes, namely, dynamic waterscape HP1 (from Phoenix Mountain National Scenic Area), static waterscape HP2 (from Green Stone Valley National Forest Park) and lookout landscape HP3 (from Phoenix Mountain National Scenic Area); three low-preference landscapes, namely, broadleaf forest landscapes LP1 (from Heyi National Forest Park), LP2 (from Heyi National Forest Park) and LP3 (from Shenyang National Forest Park). This figure was created by the author.

**Figure 4 ijerph-20-04753-f004:**
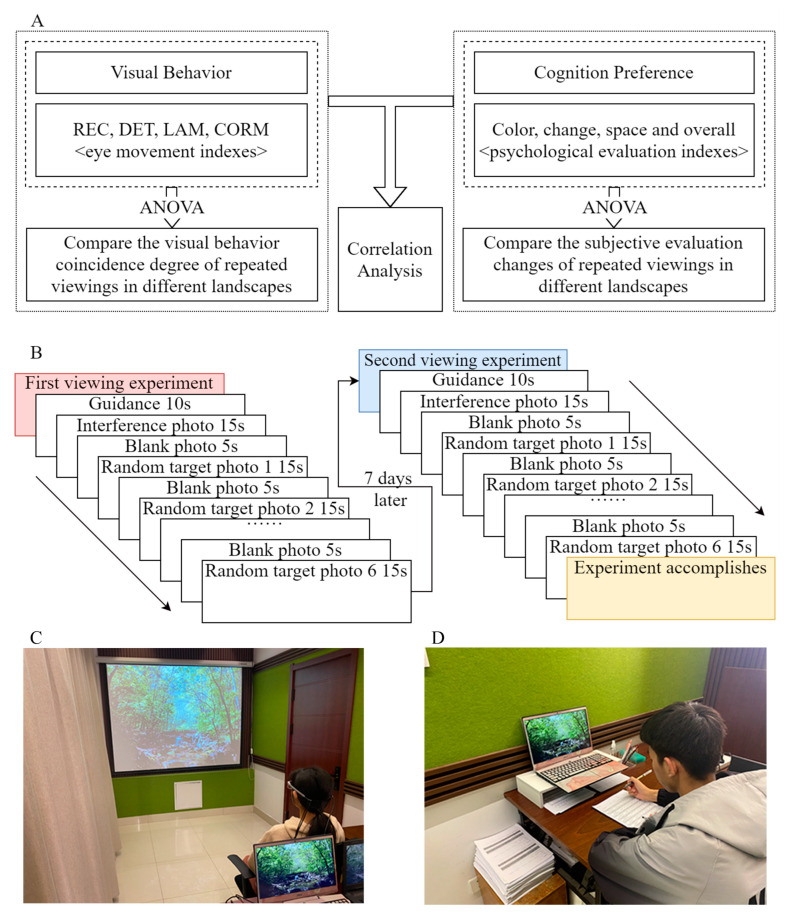
Experimental process and related experimental photos. (**A**) Overall framework of the study. (**B**) The flow of the eye movement experiment. (**C**) On-site photo of eye movement experiments. (**D**) On-site photo of psychological evaluation. The graphics software is Draw.io v13.9.9. This figure was created by the author.

**Figure 5 ijerph-20-04753-f005:**
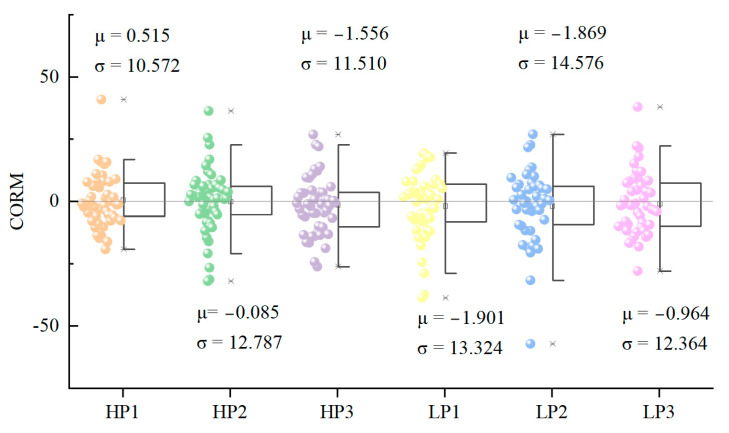
The sequence of the first and second viewings. Note: This description explains the meaning of the scope of “CORM”: when CORM > 0, the first viewing precedes the second viewing; when CORM = 0, they are basically synchronized; and when CORM < 0, the first viewing starts slower than the second one. The analysis and graphics software is Origin Pro 2021 v9.8.0.200. This figure was created by the author.

**Figure 6 ijerph-20-04753-f006:**
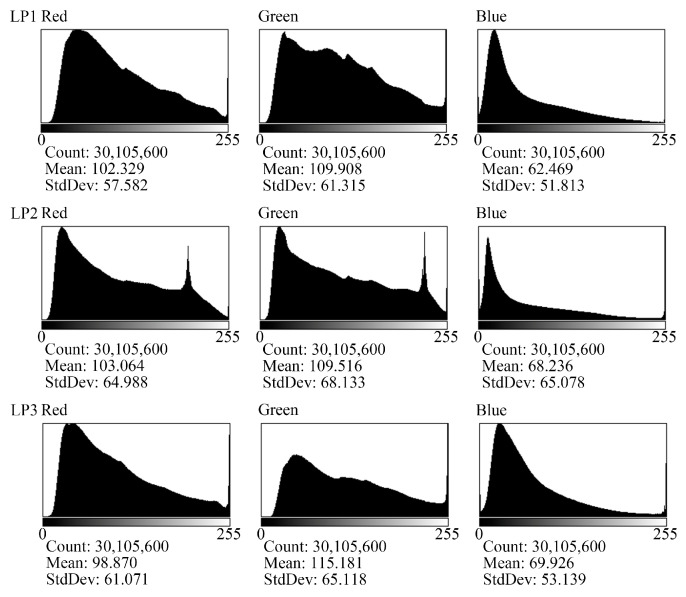
Color histogram of the low-preference landscape. Note: LP1, LP2 and LP3 refer to broadleaf forest 1, broadleaf forest 2 and broadleaf forest 3 in the low-preference landscape, respectively. The horizontal axis of the histogram represents the brightness partition of the landscape, and the vertical axis represents the number of pixels of color. The graphics software is ImageJ v1.8.0.112. This figure was created by the author.

**Figure 7 ijerph-20-04753-f007:**
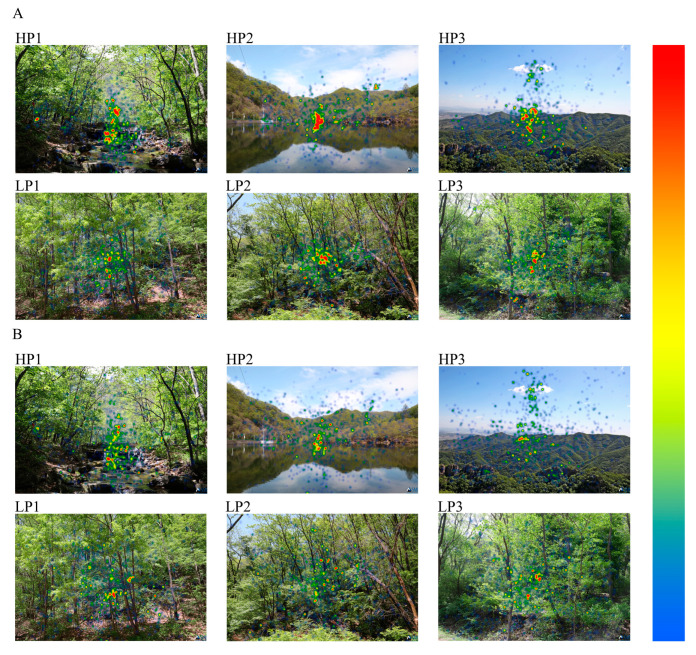
The distribution of participants’ gaze points during the first and second viewing of forest landscape spaces with different preferences ((**A**) the first viewing, (**B**) the second viewing). Note: The color gradient bar on the right represents participants’ visual attention., different colors have different meanings, the redder the color, the higher the concern, while the bluer the color, the lower the concern. This figure was created by the author.

**Figure 8 ijerph-20-04753-f008:**
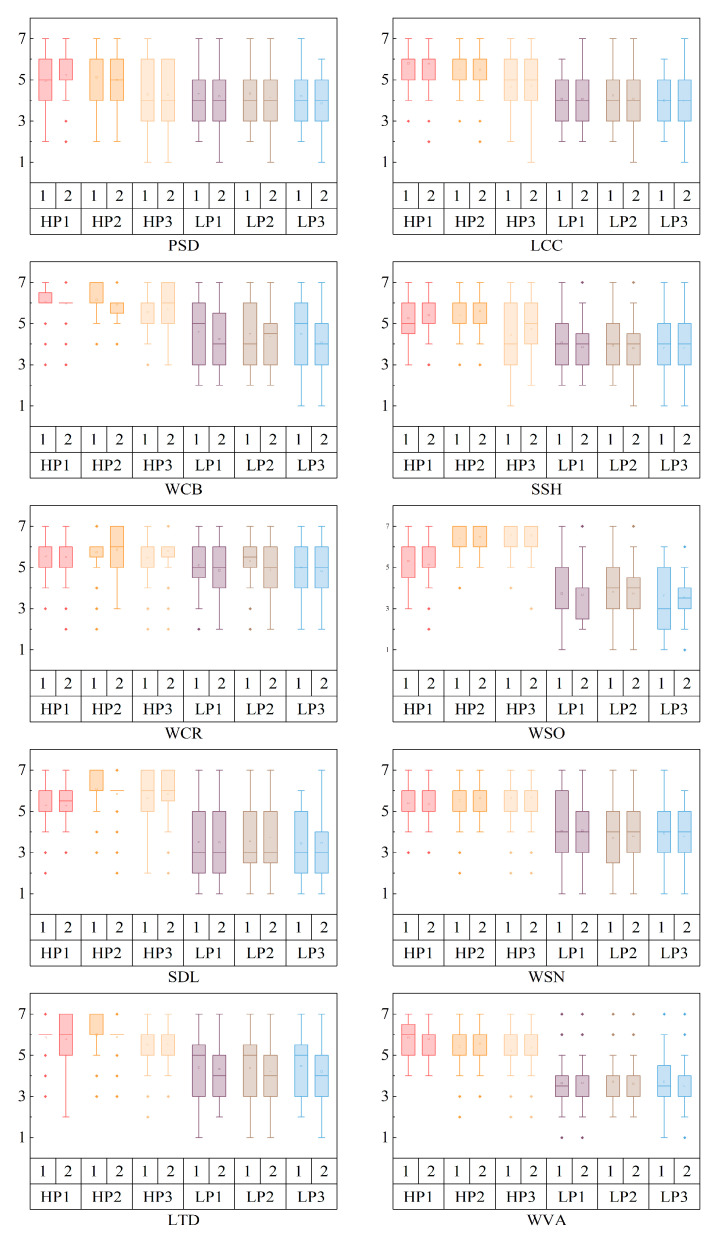
Cognitive evaluations of landscape spaces when repeatedly viewing. Note: PSD: Whether the plant species are diverse; LCC: Whether the landscape content changes; WCB: Whether the color is bright; SSH: Whether the space has a sense of hierarchy; WCR: Whether the color is rich; WSO: Whether the space is open; SDL: Whether a distant landscape can be seen; WSN: Whether the space is neat; LTD: Whether the near-middle landscape is three-dimensional; WVA: Whether the individual intends to visit again. In the horizontal axis, 1: The first viewing; 2: The second viewing. “□” stands for mean value and “•” stands for abnormal point. The analysis and graphics software are Origin Pro 2021 v9.8.0.200. This figure was created by the author.

**Figure 9 ijerph-20-04753-f009:**
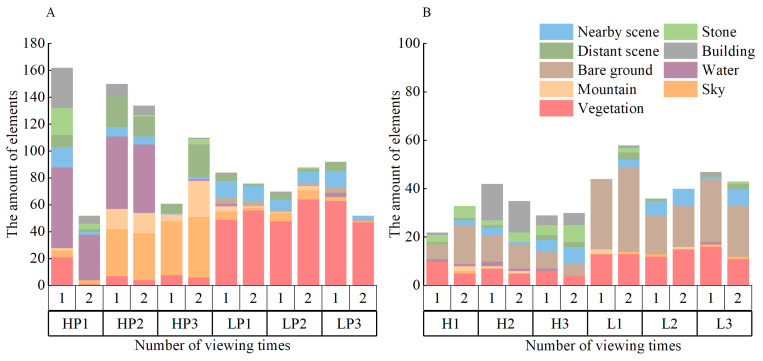
Statistics of all participants’ likes and dislikes for forest landscapes. (**A**) Statistics of likes for landscape elements; (**B**) Statistics of dislikes for landscape elements; the analysis and graphics software are Origin Pro 2021 v9.8.0.200. This figure was created by the author.

**Figure 10 ijerph-20-04753-f010:**
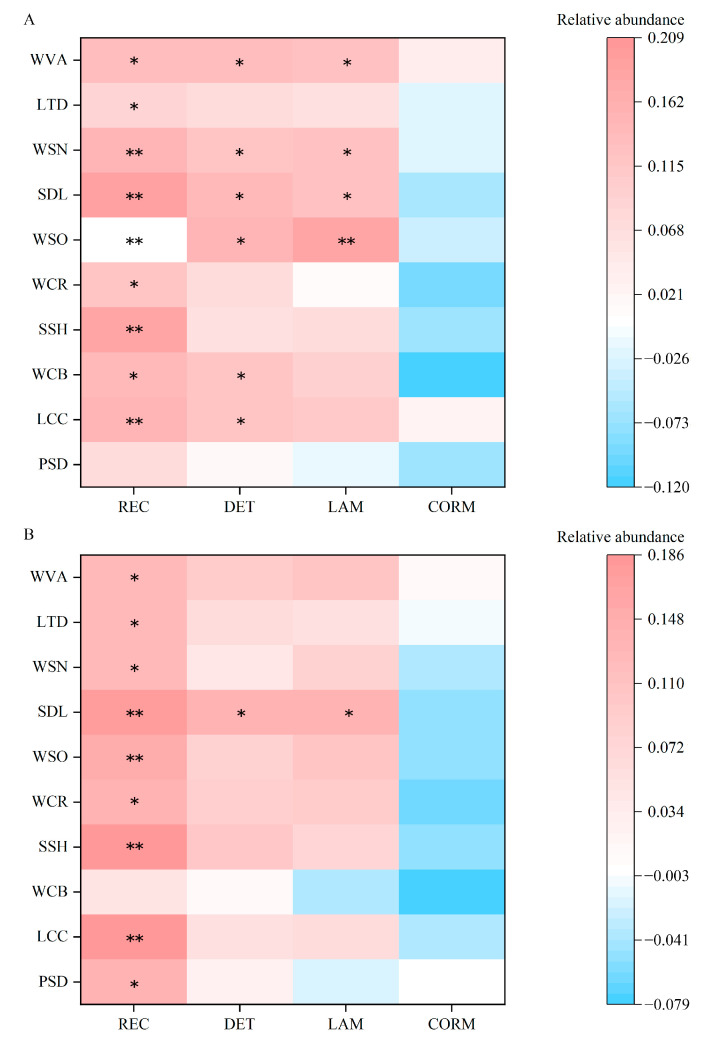
Correlation between visual behavior coincidence and cognitive evaluation. This figure was created by the author. Note: (**A**) The first viewing, (**B**) The second viewing. * significant correlation with *p* < 0.050, ** significant correlation with *p* < 0.010. The analysis and graphics software are Origin Pro 2021 v9.8.0.200. This figure was created by the author.

**Figure 11 ijerph-20-04753-f011:**
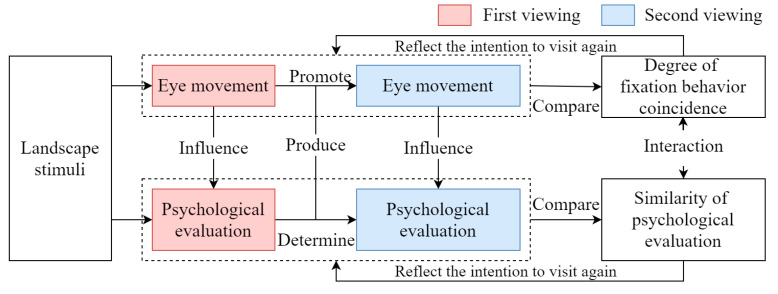
The relationship between visual behavior and landscape preference when repeatedly viewing. The graphics software is Draw.io v13.9.9. This figure was created by the author.

**Figure 12 ijerph-20-04753-f012:**
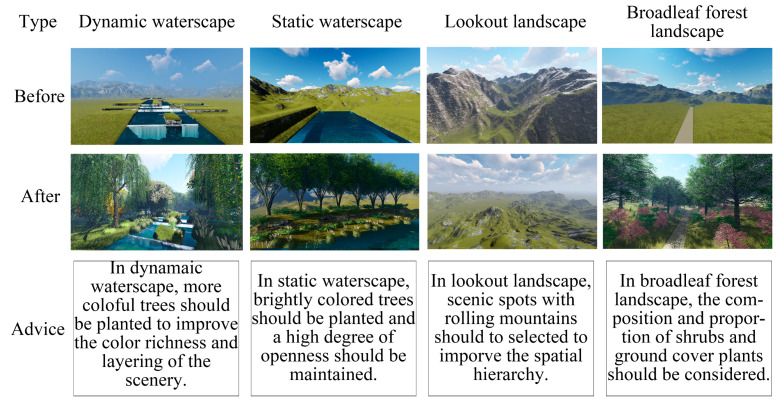
Suggestions on planning and design of forest landscapes. The graphics software is Lumion 6.0. This figure was created by the author.

**Table 1 ijerph-20-04753-t001:** CRA indexes and meanings.

Index	Algorithm Formula	Meaning
REC	$\mathrm{REC}=100\frac{2R}{N\left(N-1\right)}$	Recurrence as a percentage of all fixation points. This article refers to the coincidence degree of fixation points in repeated viewings.
DET	$\mathrm{DET}=100\frac{\left|D_{L}\right|}{R}$	The proportion of diagonal recurrences in a recurrence plot. It indicates the coincidence degree of similar segmented fixation behaviors of people in the process of repeated viewings (similar continuous fixation segments).
LAM	$\mathrm{LAM}=100\frac{\left|H_{L}\right|+\left|V_{L}\right|}{2R}$	The proportion of recurrences that are vertical or horizontal in a recurrence plot. It refers to the coincidence degree of the segmented fixations at one viewing time and the single fixation point at another viewing time.
CORM	$\mathrm{CORM}=100\frac{\sum_{N}^{N-1}\sum_{j=i+1}^{N}\;\left(j-i\right)\;r_{ij}}{\left(N-1\right)\;R}$	The distance between the main recurrence cluster in a recurrence plot and the main diagonal line indicates the interval between fixation points that gaze at the same position in repeated viewings. CORM > 0 indicates that the first viewing starts before the second viewing; CORM = 0 indicates that they are basically synchronized; and CORM < 0 indicates that the second viewing starts before the first viewing.

**Table 2 ijerph-20-04753-t002:** Index selection of landscape cognitive evaluation.

Evaluation Content	Evaluation Index	
Space	Whether the space is open	Can you see the distant landscape
	Whether the space is neat	Whether the space has a sense of hierarchy
Color	Whether the color is rich	Whether the color is bright
Landscape change	Whether the plant species are diverse	Whether the landscape content is changing
	Whether the near-middle landscape is three dimensional	
Overall	Whether to visit again	Whether you like it

**Table 3 ijerph-20-04753-t003:** Differences in the visual behavior coincidence degree across forest landscapes.

Landscape Type	REC			DET			LAM		
	Median	Lower Quartile	Upper Quartile	Median	Lower Quartile	Upper Quartile	Median	Lower Quartile	Upper Quartile
HP1 (a)	17.125 ^cdEF^	11.055	24.170	50.230 ^cdeF^	34.675	62.515	66.670 ^cdEF^	50.420	73.500
HP2 (b)	15.405 ^deF^	10.288	21.780	42.230 ^eF^	33.745	51.558	57.935 ^eF^	46.172	67.682
HP3 (c)	13.045 ^af^	8.863	17.620	38.035 ^af^	25.850	49.638	50.610 ^af^	39.938	65.560
LP1 (d)	11.495 ^ab^	7.415	16.983	38.160 ^af^	26.090	44.750	50.000 ^af^	33.325	63.695
LP2 (e)	11.910 ^Ab^	7.792	17.135	32.320 ^ab^	23.620	50.605	45.380 ^Ab^	29.045	62.115
LP3 (f)	9.615 ^ABc^	6.495	13.223	30.610 ^ABcd^	22.620	37.740	43.590 ^ABcd^	24.855	53.770

Note: REC is the coincidence degree of fixation points in repeated viewings; DET is the coincidence degree of similar segmented fixation behaviors; LAM is the coincidence degree of the segmented fixations at one viewing time and the single fixation point at another viewing time. The six stimuli are labelled a–f. When the value of one stimulus is significantly different from that of the other stimulus, *p* < 0.050 is indicated by the corresponding lowercase letter in the upper right corner, and *p* < 0.010 is indicated by the corresponding uppercase letter. Example: The REC of HP1 (a) is significantly different from that of LP3 (f), *p* < 0.010, so the value of HP1 is 17.125 ^F^.

## Data Availability

The data presented in this study are available on request from the corresponding author. The data are not publicly available due to the copyright of relevant data in the article belonging to the research group rather than to individuals.

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
