# Peer review of "How Do Repeated Viewings in Forest Landscapes Influence Young People’s Visual Behaviors and Cognitive Evaluations?"

_ijerph, 2023, doi:10.3390/ijerph20064753_

Round 1

Reviewer 1 Report

Dear authors,

the article has several merits in terms of contribution to the current debate on how to better design and manage forest landscapes. In particular, the study is of value for investigating more in depth which are the features of the landscape that can increase people’s motivation to visit forests.

The main aim of the article is to provide important insights on people’s perception and observation modes in order to facilitate the forest resource management. The main findings are based on repeated observation of the study sample.

Introduction

Lines 119-123: in this part of the article it is explained that the study aims to analyse the kind of correlation among visual behaviors that can be detected in repeated viewings. Thus, the study has manifold aim: to identify the main patterns emerging in the visual behaviors and cognitive evaluation (as introduced in the paragraph 1.2 of your article) but also to determine to which extent the first view can influence the second view and so on. Indeed, these two perspectives can go together but please consider to make the focus a bit clearer. In certain parts, including the title of the article, the main focus of the study is on how repeated viewings can influence people’s visual behaviors or cognitive evaluation. However, in other parts the analysis focuses more on identifying which features of different landscapes can attract more people, by determining a specific visual behavior and cognitive evaluation.

Line 152: is the term “psychological evaluation” used as synonymous of “cognitive evaluation”. Are there some differences in the terms that could require more explanation?

Lines 164-173: are the three hypotheses formulated for the statistical tests?Please, consider to make this aspect clearer.

Materials and methods:

As it concerns the statistical method, the choice of both test (T- Test and F- Test, i.e. ANOVA) requires more explanation. Please, make clearer if the study includes only two observations or even more than two for the same sample. In certain paragraphs it is mentioned only the two observations but in other paragraphs it refers to an increasing number of observations, but not clearly mentioned how many.

Lines 178-185: As it concerns the selection of the sample, the author states that 5 famous forests parks were identified as representative of forests located in the suburbs of cities. However, the characteristics of suburban forests can significantly change according to the geographical context. In European context, the definition of suburban forests could be within a driving distance, shorter that 1.5 hour as for the sample of the study.

Lines 214-215: the sample representativeness is based on strong assumptions that need further explanation. The assumptions of normal distribution of the sample and homogeneity of the variance within the sample have to be tested in order to support the correctness of the analysis. The sample is relatively small, but these assumptions could still be met.

Lines 214-215: moreover, it is not enough explained how the survey results of college students accord with people’s actual aesthetic preferences. A sample including elderly people could lead to different results.

Figure 8: Please, specify the software that has been used for conducting the analysis.

Lines 249-251: Could you please, explain more about the Richter scale used for the cognitive evaluation. Are you referring to the Likert scale?

Lines 253-254: could you please further explain what do you mean with “previous landscape evaluation research system”?

Lines 257 – 258 (table 2): this is meant to introduce the selected indexes for landscape cognitive evaluation. Instead of the questions, could you please describe better the indexes in itself? What is the difference between first-class and second-class? Which type of answers was proposed for these multiple-choice questions?

As a general consideration, conducting the experiments in a laboratory (i.e., with pictures) may necessitate more explanation about the potential discrepancy with observations conducted in a real environment (on the spot). Of course, this necessitates additional efforts (both financially and in terms of study organisation), which are not always feasible. This aspect, however, should be mentioned.

Results

The description of the results is very well structured but please consider to make clearer the relation with the three hypotheses (lines 164-173).

Lines 321-322: double check the sentence “for the relationship between CORM and zero”. The description explains the range of values for the index CORM.

Line 413: please, explain further how many viewings have been conducted for the study: two or more than two?

Lines 460-461: please explain better in which way you consider the analysis of correlation enough for making inference. To make inference more statistical analysis, based on larger sample, would be required.

Conclusions and suggestions

Lines: 639-640: is really the main results showing that repeated viewing can impact people’s visual behavior and cognitive assessment? Or is it more correct to pointing out that specific features in the landscape can be translated into specific visual behavior and this can influence the decision to return in the same place?

All the best for your research.

Author Response

Response letter 1

Comment 1:

the article has several merits in terms of contribution to the current debate on how to better design and manage forest landscapes. In particular, the study is of value for investigating more in depth which are the features of the landscape that can increase people’s motivation to visit forests.

The main aim of the article is to provide important insights on people’s perception and observation modes in order to facilitate the forest resource management. The main findings are based on repeated observation of the study sample.

Modify 1:

Thank you for your affirmation, and we highly cherish your comments.

Comment 2:

Lines 119-123: in this part of the article it is explained that the study aims to analyse the kind of correlation among visual behaviors that can be detected in repeated viewings. Thus, the study has manifold aim: to identify the main patterns emerging in the visual behaviors and cognitive evaluation (as introduced in the paragraph 1.2 of your article) but also to determine to which extent the first view can influence the second view and so on. Indeed, these two perspectives can go together but please consider to make the focus a bit clearer. In certain parts, including the title of the article, the main focus of the study is on how repeated viewings can influence people’s visual behaviors or cognitive evaluation. However, in other parts the analysis focuses more on identifying which features of different landscapes can attract more people, by determining a specific visual behavior and cognitive evaluation.

Modify 2:

Thanks for your advice and we value your advice very much. The purpose of this study is to explore the patterns and changing trends of people's visual behaviors and cognitive evaluations based on repeated viewing of landscapes, which are also described in the results mainly. On this basis, we want to deepen the theoretical results into concrete practice and implement them into the planning and design of forest landscapes, so we pay more attention to finding specific visual behaviors and cognitive evaluations in the part of results and in most conclusions, and determine which features of different landscapes can attract more people.

Comment 3:

Line 152: is the term “psychological evaluation” used as synonymous of “cognitive evaluation”. Are there some differences in the terms that could require more explanation?

Modify 3:

Thank you for your advice and we are sorry that we did not express it clearly. There are differences between the term “psychological evaluation” and “cognitive evaluation” in this study. We have added a definition of terms according to your advice.

See as Line 259-266. The details are as follows:

“In this study, subjective evaluation includes two terms. One is psychological evaluation and the other is cognitive evaluation.

Psychological evaluation: Including two parts, the cognitive evaluations of landscapes (including evaluations about space, color, landscape change and overall, in the form of Likert scale with scores from 1-7) and the recordings and descriptions of people's preferences for landscape elements (in the form of multiple-choice questions). It reflects the evaluations of landscape characteristics and the preference for elements.

Cognitive evaluation: It refers to the cognitive evaluations of landscapes (including evaluations about space, color, landscape change and overall, in the form of Likert scale with scores from 1-7), which only reflects the evaluation of landscape characteristics.”

Comment 4:

Lines 164-173: are the three hypotheses formulated for the statistical tests?Please, consider to make this aspect clearer.

Modify 4:

Thanks for your advice, according to your advice, we have modified this part to make the hypotheses formulated for the statistical tests.

See as Line 271-281. The details are as follows:

 “In order to reach Hypothesis 1 (H1), according to the characteristics and attributes of data, nonparametric K-W test are used to compare the differences in the visual behavioral coincidence degree of all participants when repeatedly viewing.

To reach Hypothesis 2 (H2), descriptive statistical analysis was used to compare all the landscape elements that the participants liked and disliked when they repeatedly viewed the landscapes and to clarify their changing trends with the number of times they viewed them and nonparametric Wilcoxon Signed Ranks test was used to explore the changes in psychological evaluations on the first and second viewings.

To reach Hypothesis 3 (H3), Spearman correlation analysis was used to explore the correlation between the subjective psychological evaluations and objective visual behaviors of the participants when viewing landscape stimuli.”

Comment 5:

As it concerns the statistical method, the choice of both test (T- Test and F- Test, i.e. ANOVA) requires more explanation.

Modify 5:

Thanks for your advice and we highly value it. when we first analyzed, the data was defaulted to normal distribution. According to your advice, we have checked this data again and found that the data do not obey normal distribution. Therefore, we updated the related analysis methods and carefully checked the results.

Attached is the analysis result of Normality test about visual behaviors and cognitive evaluations.

Tests of Normality about visual behaviors

Kolmogorov-Smirnova

Shapiro-Wilk

Statistic

df

Sig.

Statistic

df

Sig.

REC

0.142

285

0.000

0.827

285

0.000

LAM

0.068

285

0.003

0.982

285

0.001

DET

0.069

285

0.002

0.975

285

0.000

CORM

0.054

285

0.040

0.977

285

0.000

a. Lilliefors Significance Correction

Tests of Normality about cognitive evaluations

Kolmogorov-Smirnova

Shapiro-Wilk

Statistic

df

Sig.

Statistic

df

Sig.

PSD

0.143

620

0.000

0.942

620

0.000

LCC

0.182

620

0.000

0.919

620

0.000

WCB

0.236

620

0.000

0.893

620

0.000

SSH

0.155

620

0.000

0.937

620

0.000

WCR

0.233

620

0.000

0.888

620

0.000

WSO

0.210

620

0.000

0.900

620

0.000

SDL

0.214

620

0.000

0.902

620

0.000

WSN

0.197

620

0.000

0.918

620

0.000

LTD

0.218

620

0.000

0.905

620

0.000

WVA

0.161

620

0.000

0.934

620

0.000

a. Lilliefors Significance Correction

See as Line 289-292. The details are as follows:

“To begin with, we tested whether the data obey the assumption of normal distribution and homogeneity of variance, and found all the data do not obey normal distribution, p < 0.050, so we used K-W test to analyze the visual behavior coincidence degree. The result is shown in Table 3.”

See as Line 385-387. The details are as follows:

“We firstly tested whether the data obey the assumption of normal distribution and homogeneity of variance, and found all the data do not obey normal distribution, p < 0.050, so we used Wilcoxon test to compare the cognitive evaluations for repeated viewings.”

Comment 6:

Please, make clearer if the study includes only two observations or even more than two for the same sample. In certain paragraphs it is mentioned only the two observations but in other paragraphs it refers to an increasing number of observations, but not clearly mentioned how many.

Modify 6:

Thanks for your advice. The research in this paper only includes two viewings. But we think some results of the two-time viewing can also play a guiding role in multiple viewings, so we sometimes mention increasing number of viewings in this paper.

We are sorry that we did not express it clearly and we have modified the diction errors as far as possible from “with increase of viewing times” to “at the second viewing”, and marked the new content in red.

Comment 7:

Lines 178-185: As it concerns the selection of the sample, the author states that 5 famous forests parks were identified as representative of forests located in the suburbs of cities. However, the characteristics of suburban forests can significantly change according to the geographical context. In European context, the definition of suburban forests could be within a driving distance, shorter that 1.5 hour as for the sample of the study.

Modify 7:

Thanks for your advice. We have modified the definition of suburban forests according to your advice.

See as Line 181-182. The details are as follows:

Chang “…are within a 1.5-hour drive from the city center and thus are forest landscape resources in the suburbs of the city.”

To “…are within 60-100 km from the city center and thus are forest landscape resources in the suburbs of the city.”

Comment 8:

Lines 214-215: the sample representativeness is based on strong assumptions that need further explanation. The assumptions of normal distribution of the sample and homogeneity of the variance within the sample have to be tested in order to support the correctness of the analysis. The sample is relatively small, but these assumptions could still be met.

Modify 8:

Thanks for your advice and we highly value it. When we first analyzed, the data was defaulted to normal distribution. According to your advice, we have checked this data again and found that the data do not obey normal distribution. Therefore, we updated the related analysis methods and carefully checked the results.

Attached is the analysis result of Normality test about visual behaviors and cognitive evaluations.

Tests of Normality about visual behaviors

Kolmogorov-Smirnova

Shapiro-Wilk

Statistic

df

Sig.

Statistic

df

Sig.

REC

0.142

285

0.000

0.827

285

0.000

LAM

0.068

285

0.003

0.982

285

0.001

DET

0.069

285

0.002

0.975

285

0.000

CORM

0.054

285

0.040

0.977

285

0.000

a. Lilliefors Significance Correction

Tests of Normality about cognitive evaluations

Kolmogorov-Smirnova

Shapiro-Wilk

Statistic

df

Sig.

Statistic

df

Sig.

PSD

0.143

620

0.000

0.942

620

0.000

LCC

0.182

620

0.000

0.919

620

0.000

WCB

0.236

620

0.000

0.893

620

0.000

SSH

0.155

620

0.000

0.937

620

0.000

WCR

0.233

620

0.000

0.888

620

0.000

WSO

0.210

620

0.000

0.900

620

0.000

SDL

0.214

620

0.000

0.902

620

0.000

WSN

0.197

620

0.000

0.918

620

0.000

LTD

0.218

620

0.000

0.905

620

0.000

WVA

0.161

620

0.000

0.934

620

0.000

a. Lilliefors Significance Correction

See as Line 289-292. The details are as follows:

“To begin with, we tested whether the data obey the assumption of normal distribution and homogeneity of variance, and found all the data do not obey normal distribution, p < 0.05, so we used K-W test to analyze the visual behavior coincidence degree. The result is shown in Table 3.”

See as Line 385-387. The details are as follows:

“We firstly tested whether the data obey the assumption of normal distribution and homogeneity of variance, and found all the data do not obey normal distribution, p < 0.05, so we used Wilcoxon test to compare the cognitive evaluations for repeated viewings.”

Comment 9:

Lines 214-215: moreover, it is not enough explained how the survey results of college students accord with people’s actual aesthetic preferences. A sample including elderly people could lead to different results.

Modify 9:

Thanks for your advice,and we have corrected the diction errors: change all the words “people” to “young people”. As you suggested, a richer sample size is more convincing. The expansion of data of various demographic attributes such as age and occupation is an indispensable part in our future research, so we put this part of the description to the   limitation part of the Discussion.

See as Line 671-673. The details are as follows:

“2. On the selection of participants, this study mainly explored the visual behaviors and psychological evaluations of young people under repeated viewings. In the future research, the scope could be further expanded to people of all ages.”

Comment 10:

Figure 8: Please, specify the software that has been used for conducting the analysis.

Modify 10:

Thanks for your advice,and we have added the detailed software information below the analysis.

See as Line 404-405. The details are as follows:

“The analysis and graphics software is Origin 2021.”

Comment 11:

Lines 249-251: Could you please, explain more about the Richter scale used for the cognitive evaluation. Are you referring to the Likert scale?

Modify 11:

Thanks for your advice. We are sorry we confused these two words “Richter scale” and “Likert scale”. In fact, what we want to express is “Likert scale”, and we have already corrected the error.

See as Line 250-251. The details are as follows:

Change “…the second part is the landscape evaluation using a Richter scale (scoring from low to high is 1-7) …”

To “…the second part is the landscape evaluation using a Likert scale (scoring from low to high is 1-7) …”

See as Line 554. The details are as follows:

Change “For the cognitive evaluation part of the Richter scale…”

To “For the cognitive evaluation part of the Likert scale…”

Comment 12:

Lines 253-254: could you please further explain what do you mean with “previous landscape evaluation research system”?

Modify 12:

Thanks for your advice. We are sorry the diction we used was not suitable. The landscape evaluation index in this paper is extracted from the previous landscape evaluation system (referring to the mature indicators of previous landscape evaluations ([28, 48]), which mainly evaluates the photos from four aspects: spatial characteristics, color, landscape change and overall evaluation. According to your advice, we have changed the diction from “previous landscape evaluation research system” to “previous landscape evaluation system”.

See as Line 254-255. The details are as follows:

Change “The landscape evaluation indexes in this paper were extracted from the previous landscape evaluation research system…”

To “The landscape evaluation indexes in this study were extracted from the previous landscape evaluation system…”

Comment 13:

Lines 257 – 258 (table 2): this is meant to introduce the selected indexes for landscape cognitive evaluation. Instead of the questions, could you please describe better the indexes in itself? What is the difference between first-class and second-class? Which type of answers was proposed for these multiple-choice questions?

Modify 13:

Thanks for your advice. We are sorry the diction we used was not suitable. We have modified the terms from “first-class index” to “evaluation content” and “second-class index” to “evaluation index” to be better understood. The evaluation content includes four parts, “Space, Color, Landscape change and Overall”, and the evaluation indexes are the specific indexes of every part. The answers were collected by a Likert scale (scoring from low to high is 1-7) to know people’s views on landscapes.

See as Line 258. The details are as follows:

Evaluation content

Evaluation index

Space

Whether the space is open

Can you see the distant landscape

Whether the space is neat

Whether the space has a sense of hierarchy

Color

Whether the color is rich

Whether the color is bright

Landscape change

Whether the plant species are diverse

Whether the landscape content is changing

Whether the near-middle landscape is three dimensional

Overall

Whether to visit again

Whether you like it

Comment 14:

As a general consideration, conducting the experiments in a laboratory (i.e., with pictures) may necessitate more explanation about the potential discrepancy with observations conducted in a real environment (on the spot). Of course, this necessitates additional efforts (both financially and in terms of study organisation), which are not always feasible. This aspect, however, should be mentioned.

Modify 14:

Thanks for your advice. We added some references to explain that photos could replace on-site investigations to some extent. Nowadays, finance and data acquisition are inevitable problems for us. On-site survey would be one of our future research topics.

See as Line 199-201. The details are as follows:

“A great number of studies have explored and confirmed the relationship between on-site surveys and photos, and found that photos can replace on-site surveys to some extent. Therefore, in our research, photos were chosen to be the experimental stimuli [37,43-45].” 

At the same time, this is also a part of our future topic, and we added it to the limitation part of the Discussion.

See as Line 667-670. The details are as follows:

“And the difference between on-site surveys and photos will also have a certain impact on the experimental results. However, finance and data acquisition are inevitable problems. This would be our future research topic.”

Linking to the literature:

  1. Xiang, Y.; Liang, H.; Fang, X.; Chen, Y.; Xu, N.; Hu, M.; Chen, Q.; Mu, S.; Hedblom, M.; Qiu, L.; et al. The comparisons of on-site and off-site applications in surveys on perception of and preference for urban green spaces: Which approach is more reliable? Urban Forestry & Urban Greening 2021, 58, doi: 10.1016/j.ufug.2020.126961.
  2. Cottet, M.; Vaudor, L.; Tronchère, H.; Roux-Michollet, D.; Augendre, M.; Brault, V. Using gaze behavior to gain insights into the impacts of naturalness on city dwellers' perceptions and valuation of a landscape. Journal of Environmental Psychology 2018, 60, 9-20, doi: 10.1016/j.jenvp.2018.09.001.
  3. Dupont, L.; Ooms, K.; Duchowski, A.T.; Antrop, M.; Van Eetvelde, V. Investigating the visual exploration of the rural-urban gradient using eye-tracking. Spatial Cognition & Computation 2016, 17, 65-88, doi: 10.1080/13875868.2016.1226837.
  4. Dupont, L.; Ooms, K.; Antrop, M.; Van Eetvelde, V. Testing the validity of a saliency-based method for visual assessment of constructions in the landscape. Landscape and Urban Planning 2017, 167, 325-338, doi: 10.1016/j.landurbplan.2017.07.005.

Comment 15:

The description of the results is very well structured but please consider to make clearer the relation with the three hypotheses (lines 164-173).

Modify 15:

Thank you for your advice, and we value it very much. We added the connected introductions with the hypotheses in the Result part.

See as Line 374-376. The details are as follows:

“In summary, there is a significant difference in the similarity of visual behavior characteristics of forest landscapes after repeated viewings for high- and low-preference landscapes. Hypothesis 1 hold.”

See as Line 440-443. The details are as follows:

“Above all, there is not a significant difference in the changes in individuals’ cog-nitive evaluations of forest landscapes after repeated viewing between high- and low-preference landscapes, but the likes and dislikes of landscape elements changed at the second viewing. Hypothesis 2 does no hold.”

See as Line 488-489. The details are as follows:

“Above all, individuals’ psychological evaluations of different forest landscapes correlated with visual behaviors. Hypothesis 3 hold.”

Comment 16:

Lines 321-322: double check the sentence “for the relationship between CORM and zero”. The description explains the range of values for the index CORM.

Modify 16:

Thanks for your advice, and we value it very much. We are sorry the diction we used was not suitable. We modified the sentence according to your advice.

See as Line 338-339. The details are as follows:

Change “Note: For the relationship between CORM and 0…”

To “Note: This description explains the meaning of the scope of “CORM…”

Comment 17:

Line 413: please, explain further how many viewings have been conducted for the study: two or more than two?

Modify 17:

Thanks for your advice. We are sorry for the diction errors. The answer is two, but we may further study more than two viewings in the future. And we have changed the diction “with the increase in viewing times” to “at the second viewing” according to your advice.

See as Line 436-437. The details are as follows:

Change “With the increase in viewing times, in the low-preference stimuli, the number of likes for broadleaf forest LP3 decreases obviously…”

To “At the second viewing, in the low-preference stimuli, the number of likes for broadleaf forest LP3 decreases obviously…”

Comment 18:

Lines 460-461: please explain better in which way you consider the analysis of correlation enough for making inference. To make inference more statistical analysis, based on larger sample, would be required.

Modify 18:

Thanks for your advice and we value it very much. (1) At present, due to technical limitations and the complexity of experimental data processing, most of the researches on eye movement experiments are small and medium samples. In the future, we will try our best to obtain more scientific results from larger samples. (2) We consulted a lot of relevant literature, which explained and proved the possibility from many perspectives: previous experience towards objects allows participants to form expectations or assumptions about the objects. Based on the experience of the first viewing, people formed their own inherent understandings of objects to a certain extent, which in turn led to similar visual behaviors in the space at the second viewing. From the logic, it can be inferred that the third viewing would also be influenced by the second viewing, people would also form their own inherent understandings of objects at the second viewing so that it would influence the third viewing. All in all, we think it is a reasonable inference and assumption. At the same time, we also add relevant literature and descriptions in many places in the article to strengthen the possibility of this inference.

See as Line 108-117. The details are as follows:

“Most importantly, these studies pointed out that there are indeed interesting differences in visual behaviors when viewing repeatedly. From the psychological point of view, these differences are partly due to the increase of familiarity caused by repeated viewing. After people have deeper understandings of the objects, their viewing mode will change: people will gradually shift from the bottom-up exploration mode guided by the salience of the object to the top-down observation mode with more subjective preferences. This means it is necessary to further understand the psychological evaluations of people on the basis of exploring the changes of visual behavioral patterns when exploring the repeated viewing of forest landscapes in our study, so as to try to explore the reasons for the visual changes.”

See as Line 531-542. The details are as follows:

Menon and Levitin et al. pointed out that previous experience allows participants to form expectations or assumptions about objects. This expectation or assumption re-stricts the cognitive level of participants to objects. According to this theory, the participants had a "Peak shift" when they viewed the landscapes for the second time, that is, when they viewed the picture for the second time, based on the ex-perience of the first viewing, they had formed their own inherent understandings of the spaces to a certain extent, which in turn led to similar visual behaviors in the space at the second viewing. Because perceptual information is caused by the interaction between realistic stimulus information and memory information, and people's visual behavior is based on their perceptual information, and this is why the visual behavior at the second viewing is similar to that at the first viewing [55].

See as Line 642-646. The details are as follows:

“In addition, when individuals are in positive emotions, they will adopt more com-prehensive processing strategies, while when individuals are in negative emotions, they will adopt more specific and detailed processing strategies [66]. This also leads to significant differences in the visual behaviors of landscape spaces under different preference evaluations.”

Linking to the literature:

  1. Menon, V., & Levitin, D. A model of aesthetic appreciation and aesthetic judgments. British Journal of Psychology 2005, 95, 489−508.
  2. Forgas, J.P. Mood and judgment: The affect infusion model (AIM). Psychological bulletin 1995, 117(1), 39-66.

Comment 19:

Lines: 639-640: is really the main results showing that repeated viewing can impact people’s visual behavior and cognitive assessment? Or is it more correct to pointing out that specific features in the landscape can be translated into specific visual behavior and this can influence the decision to return in the same place?

Modify 19:

Thank you for your advice. We highly cherish your advice and have changed the expression according to your advice.

See as Line 689-691, The details are as follows:

Change “The evidence provided by our study shows that repeated viewings of forest landscapes can impact people’s visual behaviors and cognitive assessments as follows.”

To “The evidence provided by our study shows that specific features in the landscapes can be translated into specific visual behaviors and this can influence the decision to return in the same place.”

Reviewer 2 Report

This manuscript titled "How Do Repeated Viewings in Forest Landscapes Influence People's Visual Behaviors and Cognitive Evaluations?" contributes to the design and sustainable utilization of forest landscape resources in suburban areas. The methods are well described, and the results are clearly interpreted. The figures are well presented. This study is interesting and significant, but the text is not as good as the figures. Overall, the paper is well written and meets the journal's publication standard. Here are some minor suggestions I would like to make:

1. Language: The English language needs to be improved throughout the manuscript. Expression and structure are the primary problems, rather than grammar. It would be helpful if authors would restructure the text and use simple and precise language.

2. Structure: There is a need to restructure the manuscript since some sections contain a large number of mini-paragraphs, especially in the section of results. In general, it would not be appropriate to use one sentence as a paragraph, as in the examples above: L214-215, L298-300, L342-344, L375, etc.

3. Abstract: Method needs to be enhanced.

4. Introduction:This section requires improvement. These issues need to be addressed: (1) "Introduction" is so long that it is difficult to follow the main ideas. The detailed information on a single paper is not required, such as, L76-81, L112-118. (2) It is not necessary to divide the section into many subheadings. (3) Several expressions are duplicated, such as "in terms of ..." in L62 & L67 & L74.

5. Materials and Methods

L258-273: There is no need for this section to be divided into four subsections. Instead, it could be described in a single paragraph as suggested. Additionally, ANOVA was used to test difference in present study. It is necessary to provide detailed information about asymptotic normality and consistency of variance.

6. Results

6.1 L278-230: The phrase "first, second, third..." is unnecessarily formatted, also, it contains many mini-paragraphs.

6.2 Figure 8: What’s the meaning of “1” and “2” in the horizontal axis? Define when it first appears.

7. Discussion: The width is acceptable, but the depth requires improvement. A systematic interpretation of the reasons why repeated viewings in forest landscapes affect people's visual behaviors and cognitive evaluations is needed to be presented in this section. Adding more literature will enhance this discussion.

8. Others:

8.1 All 59 participants are young undergraduates and postgraduates. I suggest replacing the word "people's" in the title with "young people's". This is only a suggestion, not a criticism.

8.2 In L216, L629, L635, etc., the phrase "in this paper" should be corrected to "in this study".

Author Response

Response letter 2

Comment 1:

This manuscript titled "How Do Repeated Viewings in Forest Landscapes Influence People's Visual Behaviors and Cognitive Evaluations?" contributes to the design and sustainable utilization of forest landscape resources in suburban areas. The methods are well described, and the results are clearly interpreted. The figures are well presented. This study is interesting and significant, but the text is not as good as the figures. Overall, the paper is well written and meets the journal's publication standard.

Modify 1:

Thank you for your affirmation and we highly cherish your comments.

Comment 2:

  1. Language: The English language needs to be improved throughout the manuscript. Expression and structure are the primary problems, rather than grammar. It would be helpful if authors would restructure the text and use simple and precise language.

Modify 2:

Thank you for your advice. We cherish your advice. Following your advice, we reorganize the language in the manuscript as far as possible and marked the new content in red.

Comment 3:

  1. Structure: There is a need to restructure the manuscript since some sections contain a large number of mini-paragraphs, especially in the section of results. In general, it would not be appropriate to use one sentence as a paragraph, as in the examples above: L214-215, L298-300, L342-344, L375, etc.

Modify 3:

Thank you for your advice. According to your advice, we have integrated a large number of mini-paragraphs. Because the modifications are too many, we have marked the relevant content in red: L217-221, L291-292, L293-316, L360-362, L397.

Comment 4:

  1. Abstract: Method needs to be enhanced.

Modify 4:

Thanks for your advice and we highly value it. When we first analyzed, the data was defaulted to normal distribution. According to your advice, we have checked this data again and found that the data do not obey normal distribution. Therefore, we updated the related analysis methods and carefully checked the results.

Attached is the analysis result of Normality test about visual behaviors and cognitive evaluations.

Tests of Normality about visual behaviors

Kolmogorov-Smirnova

Shapiro-Wilk

Statistic

df

Sig.

Statistic

df

Sig.

REC

0.142

285

0.000

0.827

285

0.000

LAM

0.068

285

0.003

0.982

285

0.001

DET

0.069

285

0.002

0.975

285

0.000

CORM

0.054

285

0.040

0.977

285

0.000

a. Lilliefors Significance Correction

Tests of Normality about cognitive evaluations

Kolmogorov-Smirnova

Shapiro-Wilk

Statistic

df

Sig.

Statistic

df

Sig.

PSD

0.143

620

0.000

0.942

620

0.000

LCC

0.182

620

0.000

0.919

620

0.000

WCB

0.236

620

0.000

0.893

620

0.000

SSH

0.155

620

0.000

0.937

620

0.000

WCR

0.233

620

0.000

0.888

620

0.000

WSO

0.210

620

0.000

0.900

620

0.000

SDL

0.214

620

0.000

0.902

620

0.000

WSN

0.197

620

0.000

0.918

620

0.000

LTD

0.218

620

0.000

0.905

620

0.000

WVA

0.161

620

0.000

0.934

620

0.000

a. Lilliefors Significance Correction

we have modified the methods properly and described them more accurately.

See as Line 18-22. The details are as follows:

Chang “This study collected data from 52 graduate and undergraduate students and analyzed the coincidence of visual behavior and preference evaluation by ANOVA, paired sample T test and Pearson correlation analysis.”

To “We used difference test to compare the differences of the visual behavior coincidence degree and the changes in psychological evaluations, descriptive statistical analysis to explore young peoples’ likes and dislikes of landscape elements, and Spearman correlation analysis to explore the correlation between the psychological evaluations and visual behaviors.”

Comment 5:

  1. Introduction:This section requires improvement. These issues need to be addressed: (1) "Introduction" is so long that it is difficult to follow the main ideas. The detailed information on a single paper is not required, such as, L76-81, L112-118. (2) Itis not necessary to divide the section into many subheadings. (3) Several expressions are duplicated, such as "in terms of ..." in L62 & L67 & L74.

Modify 5:

Thanks for your advice, according to your advice, (1) we have deleted some single paper details. (2) we have deleted the unnecessary subheadings. (3) we have modified the redundancy in the text.

See as Line 75-79. The details are as follows:

“As for seasonal changes and regional landscape features, Paraskevopoulou et al. found that deciduous trees played a positive role in rehabilitation therapy [22]. Zhang et al. found that designs with regional features had positive significance for urban landscape protection and reconstruction [23]. Millar et al. found that relatively undeveloped and agricultural landscapes attract the dynamic attention of participants [24].”

See as Line 104-108. The details are as follows:

“For example, Humphrey and Underwood, David Noton, Zeni and Hannah Faye Chua et.al studied from different perspectives, aimed at exploring people’s visual behavior characteristics and construct related schemas on repeatedly viewings, and searching for the reasons why people' s visual behaviors alike,where people put more attention when repeatedly viewing and how they distribute their attention [29,31-33].”

Comment 6:

L258-273: There is no need for this section to be divided into four subsections. Instead, it could be described in a single paragraph as suggested. Additionally, ANOVA was used to test difference in present study. It is necessary to provide detailed information about asymptotic normality and consistency of variance.

Modify 6:

Thanks for your advice, and we have considered the relationship between statistical tests and hypothesis. At the same time, we have also considered the advice of reviewer 1 that the analysis methods should be related to the hypotheses. Therefore, we try to modify this part by the following way.

See as Line 271-281. The details are as follows:

“In order to reach Hypothesis 1 (H1), according to the characteristics and attributes of data, nonparametric K-W test are used to compare the differences in the visual behavioral coincidence degree of all participants when repeatedly viewing.

To reach Hypothesis 2 (H2), descriptive statistical analysis was used to compare all the landscape elements that the participants liked and disliked when they repeatedly viewed the landscapes and to clarify their changing trends with the number of times they viewed them and Wilcoxon Signed Ranks test was used to explore the changes in psychological evaluations on the first and second viewings.

To reach Hypothesis 3 (H3), Spearman correlation analysis was used to explore the correlation between the subjective psychological evaluations and objective visual behaviors of the participants when viewing landscape stimuli.”

When we first analyzed, the data was defaulted to normal distribution. According to your advice, we have checked this data again and found that the data do not obey normal distribution. Therefore, we updated the related analysis methods and carefully checked the results.

Tests of Normality about visual behaviors

Kolmogorov-Smirnova

Shapiro-Wilk

Statistic

df

Sig.

Statistic

df

Sig.

REC

0.142

285

0.000

0.827

285

0.000

LAM

0.068

285

0.003

0.982

285

0.001

DET

0.069

285

0.002

0.975

285

0.000

CORM

0.054

285

0.040

0.977

285

0.000

a. Lilliefors Significance Correction

Tests of Normality about cognitive evaluations

Kolmogorov-Smirnova

Shapiro-Wilk

Statistic

df

Sig.

Statistic

df

Sig.

PSD

0.143

620

0.000

0.942

620

0.000

LCC

0.182

620

0.000

0.919

620

0.000

WCB

0.236

620

0.000

0.893

620

0.000

SSH

0.155

620

0.000

0.937

620

0.000

WCR

0.233

620

0.000

0.888

620

0.000

WSO

0.210

620

0.000

0.900

620

0.000

SDL

0.214

620

0.000

0.902

620

0.000

WSN

0.197

620

0.000

0.918

620

0.000

LTD

0.218

620

0.000

0.905

620

0.000

WVA

0.161

620

0.000

0.934

620

0.000

a. Lilliefors Significance Correction

we have modified the methods properly and examined the results carefully.

See as Line 289-292. The details are as follows:

“To begin with, we tested whether the data obey the assumption of normal distribution and homogeneity of variance, and found all the data do not obey normal distribution, p < 0.05, so we used K-W test to analyze the visual behavior coincidence degree. The result is shown in Table 3.”

See as Line 385-387. The details are as follows:

“We firstly tested whether the data obey the assumption of normal distribution and homogeneity of variance, and found all the data do not obey normal distribution, p < 0.05, so we used Wilcoxon test to compare the cognitive evaluations for repeated viewings.”

Comment 7:

6.1 L278-230: The phrase "first, second, third..." is unnecessarily formatted, also, it contains many mini-paragraphs.

Modify 7:

Thank you for your advice. According to your advice, we have deleted unnecessary format and integrated mini-paragraphs.

See as Line 293-316. The details are as follows:

 “The recurrence (REC) of the dynamic waterscape HP1 (17.125) with high preference was higher than that of HP3 (13.045) and LP1 (11.495), p < 0.050, than that of LP2(11.910) and LP3(9.615), p < 0.001. In addition, the static waterscape HP2 (15.405) with high preference was higher than all landscapes with low preference for LP1 (11.495), LP2(11.910) and LP3(9.615). And HP3(13.045) was higher than that of LP3(9.615). There was no significant difference among the other landscapes (p > 0.050). The recurrence of all six stimuli in repeated viewings was between 10% and 20%, which means that the coincidence degrees of the fixation points in repeated viewings were generally low, and the participants were more inclined to view the areas they had not seen in the first viewing. The determinism (DET) of the dynamic waterscape HP1 (50.230) was higher than that of the lookout landscape HP3 (38.035) and all landscapes with low preferences for LP1 (38.160), LP2 (32.320), p < 0.050 and LP3 (30.610), p < 0.001. On the other hand, the determinism of the static waterscape HP2 (42.230) was significantly higher than that of LP2(32.320), p < 0.050 and LP3 (30.610), p < 0.001. Besides, the determinism of HP3(38.035) and LP1(38.160) were higher than that of LP3(30.610), p < 0.050. There was no significant difference among the other landscapes (p > 0.050). And the laminarity (LAM) of the dynamic waterscape HP1 (66.670) was significantly higher than that of the lookout landscape HP3 (50.610) and all landscapes with low preferences LP1 (50.000), p < 0.050, LP2 (45.380) and LP3 (43.590), p < 0.001. On the other hand, the laminarity of the static waterscape HP2 (57.935) was significantly higher than those of LP2 (47.102) and LP3 (40.748). In addition, LP3 (40.748) was significantly lower than all those with high preferences, and it was also significantly lower than LP1 (50.000). The mean values and medians of DET and LAM were both in the range of 30%~70%, which means that in repeated viewings, the participants exhibited similar visual behaviors when viewing the same fixation points.”

Comment 8:

6.2 Figure 8: What’s the meaning of “1” and “2” in the horizontal axis? Define when it first appears.

Modify 8:

Thank you for your advice and we highly cherish your comments. According to your advice, we have added an explanation about “1” and “2” in the horizontal axis below the Figure 8.

See as Line 404. The details are as follows:

“…In the horizontal axis, 1: The first viewing; 2: The second viewing.”

Comment 9:

  1. Discussion: The width is acceptable, but the depth requires improvement. A systematic interpretation of the reasons why repeated viewings in forest landscapes affect people's visual behaviors and cognitive evaluations is needed to be presented in this section. Adding more literature will enhance this discussion.

Modify 9:

Thanks for your review, and we highly cherish your advice. Previous studies on repeated viewing did not set foot in the field of landscape, but mostly from the perspective of psychology and physics. We mentioned meaningful references in the introduction before, but we didn't summarize the reasons why repeated viewing of forest landscape affected people's visual behaviors and cognitive evaluations. According to the structure of the article, we try to add this explanation to the introduction, so as to better lead to this topic. And we have also added some literature and descriptions in the Discussion part.

See as Line 108-117. The details are as follows:

“Most importantly, these studies pointed out that there are indeed interesting differences in visual behaviors when viewing repeatedly. From the psychological point of view, these differences are partly due to the increase of familiarity caused by repeated viewing. After people have deeper understandings of the objects, their viewing mode will change: people will gradually shift from the bottom-up exploration mode guided by the salience of the object to the top-down observation mode with more subjective preferences. This means it is necessary to further understand the psychological evaluations of people on the basis of exploring the changes of visual behavioral patterns when exploring the repeated viewing of forest landscapes in our study, so as to try to explore the reasons for the visual changes.”

See additions in the Discussion part as Line 531-542. The details are as follows:

“Menon and Levitin et al. pointed out that previous experience allows participants to form expectations or assumptions about objects. This expectation or assumption re-stricts the cognitive level of participants to objects. According to this theory, the participants had a "Peak shift" when they viewed the landscapes for the second time, that is, when they viewed the picture for the second time, based on the ex-perience of the first viewing, they had formed their own inherent understandings of the spaces to a certain extent, which in turn led to similar visual behaviors in the space at the second viewing. Because perceptual information is caused by the interaction between realistic stimulus information and memory information, and people's visual behavior is based on their perceptual information, and this is why the visual behavior at the second viewing is similar to that at the first viewing [55].”

See as Line 642-646. The details are as follows:

“In addition, when individuals are in positive emotions, they will adopt more comprehensive processing strategies, while when individuals are in negative emotions, they will adopt more specific and detailed processing strategies [66]. This also leads to significant differences in the visual behaviors of landscape spaces under different preference evaluations.”

Linking to the literature:

  1. Menon, V., & Levitin, D. A model of aesthetic appreciation and aesthetic judgments. British Journal of Psychology 2005, 95, 489−508.
  2. Forgas, J.P. Mood and judgment: The affect infusion model (AIM). Psychological bulletin 1995, 117(1), 39-66.

Comment 10:

8.1 All 59 participants are young undergraduates and postgraduates. I suggest replacing the word "people's" in the title with "young people's". This is only a suggestion, not a criticism.

Modify 10:

Thanks for your review, and we highly cherish your advice. We have modified the title according to your advice, and all the related words in the article. Besides, we added a relevant explanation to better introduce the Participants part.

See as Line 196-198. The details are as follows:

“In addition, due to the age limit of the participants, in this study, we explore young people’s visual behaviors in spring when green plants grow.”

Comment 11:

8.2 In L216, L629, L635, etc., the phrase "in this paper" should be corrected to "in this study".

Modify 11:

Thanks for your advice and we highly cherish it. We have modified all the diction errors from “in this paper” to “in this study” according to your advice, and marked the new content in red.

Reviewer 3 Report

The paper investigates how the repeated observation of different forest landscapes affects the visual behavior and cognitive evaluations of people.

The work presents an exciting topic, significantly enhanced by the impact of COVID-19 and reduced going out due to the lockdown, after which the need to go out into nature became more important. research on this topic is extremely important for further communication and the work of landscape architects who can adequately approach landscape design.

The work is interestingly described, and the hypotheses, methods, and results are clearly presented.

However, I have a couple of comments that I would like to highlight so that the authors can think about them in order to improve the work:

2.2. Stimulus: from which region (from picture 2) does the individual photo from picture 3 originate

2.3. Participants: consider whether valid data including 52 participants is your result. Namely, initially, you had 59 participants, and during the research, your results showed several invalid answers, so you got 52 participants. I suggest you move lines 217-221 to the Results. In this chapter, describe in more detail the age of the respondent, age, if you know the origin (where they are from), etc.

Discussion, conclusions, and further suggestions are very well indicated. This work provides good guidelines for the way of perceiving the environment and can be of great help to landscape architects in future planning.

Author Response

Response letter 3

Comment 1:

The paper investigates how the repeated observation of different forest landscapes affects the visual behavior and cognitive evaluations of people.

The work presents an exciting topic, significantly enhanced by the impact of COVID-19 and reduced going out due to the lockdown, after which the need to go out into nature became more important. research on this topic is extremely important for further communication and the work of landscape architects who can adequately approach landscape design.

The work is interestingly described, and the hypotheses, methods, and results are clearly presented.

Modify 1:

Thank you for your affirmation, and we highly cherish your comments.

Comment 2:

2.2. Stimulus: from which region (from picture 2) does the individual photo from picture 3 originate.

Modify 2:

Thanks for your advice. We have added the origin introductions according to your advice.

See as Line 210-215. The details are as follows:

Chang “…dynamic waterscape HP1, static waterscape HP2 and lookout landscape HP3; Three low-preference landscapes namely broadleaf forest landscapes LP1, LP2 and LP3.”

To “…dynamic waterscape HP1 (from Phoenix Mountain National Scenic Area), static waterscape HP2 (from Green Stone Valley National Forest Park) and lookout landscape HP3 (from Phoenix Mountain National Scenic Area); Three low-preference landscapes namely broadleaf forest landscapes LP1 (from Heyi National Forest Park), LP2 (from Heyi National Forest Park) and LP3 (from Shenyang National Forest Park).”

Comment 3:

2.3. Participants: consider whether valid data including 52 participants is your result. Namely, initially, you had 59 participants, and during the research, your results showed several invalid answers, so you got 52 participants. I suggest you move lines 217-221 to the Results. In this chapter, describe in more detail the age of the respondent, age, if you know the origin (where they are from), etc.

Modify 3: 

Thanks for your advice. We have moved lines 218-221 to the Results part according to your advice. And we added all the participants’ origin in this chapter. But considering the load caused by the questionnaire, we didn't collect more demographic attributes in the experiment.

See as Line 283-287. The details are as follows:

“3. Results

In this study, 59 undergraduates and postgraduates from Shenyang Agricultural University took part in the experiment, and those who did not participate in the second experiment (3), those whose eye movement results could not be derived (3) and those whose data were invalid (1) were screened out. At last, the data of 52 participants were valid in all.

3.1…”

Comment 4:

Discussion, conclusions, and further suggestions are very well indicated. This work provides good guidelines for the way of perceiving the environment and can be of great help to landscape architects in future planning.

Modify 4: 

Thank you for your affirmation, and we highly cherish your comments.
